# Engineering potent live attenuated coronavirus vaccines by targeted inactivation of the immune evasive viral deubiquitinase

Sebenzile K. Myeni [1] ✉, Peter J. Bredenbeek[1], Robert C. M. Knaap[1], Tim J. Dalebout[1], Shessy Torres Morales[1], Igor A. Sidorov[1], Marissa E. Linger[1], Nadia Oreshkova[1], Sophie van Zanen-Gerhardt[2], Serge A. L. Zander [2], Luis Enjuanes [3], Isabel Sola [3], Eric J. Snijder [1] & Marjolein Kikkert [1] ✉

Coronaviruses express a papain-like protease (PLpro) that is required for replicase polyprotein maturation and also serves as a deubiquitinating enzyme (DUB). In this study, using a Middle East respiratory syndrome virus (MERS-CoV) PLpro modified virus in which the DUB is selectively inactivated, we show that the PLpro DUB is an important MERS-CoV interferon antagonist and virulence factor. Although the DUB-negative rMERS-CoV$_{MA}$ replicates robustly in the lungs of human dipeptidyl peptidase 4 knock-in (hDPP4 KI) mice, it does not cause clinical symptoms. Interestingly, a single intranasal vaccination with DUB-negative rMERS-CoV$_{MA}$ induces strong and sustained neutralizing antibody responses and sterilizing immunity after a lethal wt virus challenge. The survival of naïve animals also significantly increases when sera from animals vaccinated with the DUB-negative rMERS-CoV$_{MA}$ are passively transferred, prior to receiving a lethal virus dose. These data demonstrate that DUB-negative coronaviruses could be the basis of effective modified live attenuated vaccines.

Middle East respiratory syndrome coronavirus (MERS-CoV) was first reported as a cause of severe pneumonia in humans in Saudi Arabia in 2012. Since then, the virus has been reported in 27 different countries, resulting in over 900 deaths with a high case fatality rate of around 35%[1]. MERS-CoV is widespread among dromedary camels[2–4], which form a source of animal-to-human transmission in high-risk countries like Saudi Arabia. Unlike the severe acute respiratory syndrome-coronavirus-2 (SARS-CoV-2), which is highly transmissible between humans[5], MERS-CoV is transmitted at a low frequency between humans but continues to cause localized outbreaks resulting in severe disease[1,6]. While several vaccine candidates[7], including one based on DNA[8] and three based on viral vectors[9–11], are under development for MERS-CoV at this time, none of them have been approved for use in humans. The continuous replication of MERS-CoV in animals and humans however carries the risk of the appearance of variants that may become better transmissible between humans, thereby posing the risk of another pandemic coronavirus outbreak. This highlights the need for the development of an effective MERS-CoV vaccine to control this pathogen. Additionally, since the current COVID-19 vaccines generally do not prevent (re-)infection, there may be room for alternative coronavirus vaccine designs providing more effectiveness.

Like SARS-CoV and SARS-CoV-2, the agents causing severe acute respiratory syndrome and Covid-19, respectively, MERS-CoV is an enveloped, single-stranded, positive-sense RNA virus belonging to the *betacoronavirus* genus of the *Coronaviridae* family[12,13]. MERS-CoV encodes two cysteine proteases required for proteolytic maturation of its replicase polyproteins, the papain-like protease (PLpro,

[1]Molecular Virology Laboratory, Department of Medical Microbiology, Leiden University Medical Center, Leiden, the Netherlands. [2]Experimental Pathology Services Lab, Central Laboratory Animal Facility, Leiden University Medical Center, Leiden, The Netherlands. [3]Department of Molecular and Cell Biology, National Center of Biotechnology (CNB-CSIC), Campus Universidad Autonoma de Madrid, Madrid, Spain. ✉e-mail: s.k.myeni@lumc.nl; m.kikkert@lumc.nl

embedded in nsp3) and the 3C-like or "main" protease (3CLpro or Mpro, present within nsp5). PLpro cleaves three sites in the nsp1-nsp4 region of the replicase, while Mpro releases the remaining twelve non-structural proteins. These replicase cleavage products have different roles in viral RNA synthesis and pathogenesis[12,14]. Viral proteases can have additional functions, for example by antagonizing host innate immune responses that are mounted, initially as an inflammatory response and eventually as an interferon response to efficiently counteract viral infections[15]. In turn, viral proteins including proteases can undermine these responses by actively interfering with the host's interferon induction and/or signaling. Upon infection, type I (IFN-I), and type III (IFN-λ) interferon production are triggered following the recognition by host sensors of pathogen-associated molecular patterns (PAMPs), such as double-stranded (ds) RNA[16–18]. IFN-I induces the expression of hundreds of interferon-stimulated genes (ISGs) to generate an antiviral state in host cells that limits virus replication and spread[19]. Together with other downstream effectors controlled by IFN-I (including other pro-inflammatory cytokines), ISGs also enhance pathogen recognition and thereby innate immune signaling. This also causes the recruitment and activation of various immune cells, thereby triggering a prolonged adaptive antiviral immune response[20]. A well-regulated, localized, and robust innate immune response is crucial and therefore the activities of signaling molecules in these pathways are tightly regulated by post-translational modifications such as ubiquitination[21]. For example, ubiquitination is essential for the activation or induced degradation of many factors in the signaling cascade, while specific deubiquitinating enzymes (DUBs) can downregulate the signaling to protect cells from adverse over-reactions[21–24]. The PLpro proteases of multiple coronaviruses, including MERS-CoV, possess deubiquitinase (DUB) and deISGylation (deconjugating interferon-stimulated gene 15 (ISG15)) activities[25], which serve to antagonize ubiquitin and ubiquitin-like modifications like ISG15, thus dampening inflammation and antiviral signaling[22].

Targeted modulation of virus-encoded IFN-antagonists can be an approach for the development of modified live virus (MLV) vaccines. As during natural infection, such modified but viable viruses are capable of inducing protective humoral and cellular immune responses, but since they lack (part of) the virus' innate immune suppressive activities this potentially benefits the immune response as a whole, and attenuates the modified virus. This concept has been successfully applied to a range of viruses, including poxviruses[26], flaviviruses[27], and the more advanced studies of influenza viruses lacking the non-structural protein 1 (NS1)[28] which are under clinical development[29]. Previously, the in vivo attenuation of MERS-CoV driven by augmented IFN-β and/or INF-λ host responses was achieved by deletion of open reading frames (ORF) 3 to 5[30], or by the inactivation of the nsp16 2'-O-methyltransferase[31]. Using over-expression in cell culture systems, we and others described amino acid substitutions in the ubiquitin-binding site of MERS-CoV PLpro that specifically disrupt its DUB activity without affecting the overall proteolytic activity that is crucial for viral replication. These DUB-negative modified viruses lost the ability to suppress the activation of the IFN-β/NF-κB promoter[32].

In this study, we removed the DUB activity of MERS-CoV PLpro in the context of the complete virus and evaluated its role during replication in cell culture as well as in a lethal mouse model of infection. We established that the DUB-negative rMERS-CoV$_{MA}$ is an attenuated but replication-competent virus, and therefore a potential potent MERS MLV vaccine candidate. We show that a single intranasal dose with the attenuated live DUB-negative rMERS-CoV$_{MA}$ can induce a robust neutralizing antibody response, offer complete protection against a lethal MERS-CoV challenge and provide sterilizing immunity. We also show that the protection induced by DUB-negative rMERS-CoV$_{MA}$ is largely based on antibodies, as passive transfer of immune sera to naive mice also limited MERS-CoV infection and provided a clear survival

advantage against a lethal rMERS-CoV$_{MA}$ challenge. Collectively, this study provides a proof-of-concept for the design and further development of MLV coronavirus vaccines based on the selective inactivation of their PLpro DUB activity.

## Results

### DUB-negative MERS-CoV is replication-competent in vitro and in hDPP4 KI mice

We previously characterized a panel of amino acid substitutions in the Ub-binding site of MERS-CoV PLpro that specifically disrupt its DUB activity without affecting the overall proteolytic activity. Using ectopic expression of mutant PLpro domains, we could implicate the DUB activity in antagonizing the host innate immune response. A PLpro mutant in which one amino acid was substituted at position 1691 (V1691R), resulted in severely impaired DUB activity and reduced IFN-β promoter inhibition[32]. Here, we aimed to establish the importance of PLpro DUB activity in the context of MERS-CoV infection in vitro and in vivo. To this end, we used a bacterial artificial chromosome (BAC)-based MERS-CoV reverse genetics system[33], based on the sequences of the EMC/2012 isolate[34], to produce recombinant MERS-CoVs with the V1691R amino acid substitution for in vitro studies and a mouse-adapted version of the virus for in vivo studies[35,36]. The replication kinetics of the wild-type (wt) rMERS-CoV and the rMERS-CoV$_{DUBneg}$ (DUB-negative MERS-CoV) were first analyzed in Huh7 and MRC5 cells. The DUB-negative rMERS-CoV showed essentially identical replication kinetics as wt rMERS-CoV (Fig. 1a, b). Like their MERS-CoV EMC/2012 counterparts, the two mouse-adapted viruses, rMERS-CoV$_{MA}$ and DUB-negative rMERS-CoV$_{MA}$, grew to similar titers by 24 h post infection in Huh7 and MRC5 cells infected at MOI 1 (Supplementary Fig. 1A, B). To verify whether the substitution (V1691R) was retained during virus propagation in Huh7 and MRC5 cells, RNA was isolated from the rMERS-CoV and the DUB-negative rMERS-CoV virus-containing supernatants and used for RT-PCR amplification and Sanger sequencing. The PLpro-coding region of the genome was sequenced and after five passages the presence of the engineered substitution (V1691R) was confirmed, in the absence of any additional (unintended) substitutions (Supplementary Fig. 2A)). In order to obtain a more detailed assessment of the genetic stability of the mutant virus in the substitution region or elsewhere in the genome, we passaged both rMERS-CoV and rMERS-CoV$_{DUBneg}$ ten times in HuH7 cells using a low MOI of 0.01 and virus harvest at 56hpi, after which the full genome sequences were analyzed using next-generation sequencing (NGS). A summary of the results is shown in Supplementary Table S1. After ten passages, 62% of the reads at ORF1a codon 1691/838 still contained the originally introduced V-to-R mutation, while variations encoding C, S, and H (all requiring a single-nucleotide substitution) were observed with similar low frequencies of 8–12%. Reversions to the wild-type V residue (requiring a double nucleotide substitution) could not be detected. This demonstrated that modified rMERS-CoV carrying a change in the Ub-binding site of PLpro was viable and reasonably stable in cell culture.

To further characterize the DUB-negative rMERS-CoV$_{MA}$ in vivo, we utilized the previously described lethal human dipeptidyl peptidase 4 knock-in (hDPP4 KI) mouse model for rMERS-CoV$_{MA}$ infection[35,36]. Groups of both female and male hDPP4 KI mice ($n = 10$ per group) were infected with a wt lethal dose of $10^4$ PFU of the rMERS-CoV$_{MA}$[36] or DUB-negative rMERS-CoV$_{MA}$ virus, or mock-infected with DMEM. In one experiment, animals from each group were sacrificed at day 1 and 3 p.i. and in another experiment at day 2, 4, 6, and 14. Lung virus titers in mice infected with DUB-negative rMERS-CoV$_{MA}$ were significantly (~1.5 log) lower when compared with those infected with the parental rMERS-CoV$_{MA}$ virus at day 1 and 4 p.i. (Fig. 1c, d). Of note, even though the DUB-negative rMERS-CoV$_{MA}$ progeny titers were lower than those of the parental strain, they were ~1–2 × 10^6 PFU per g of lung tissue during the first four days after

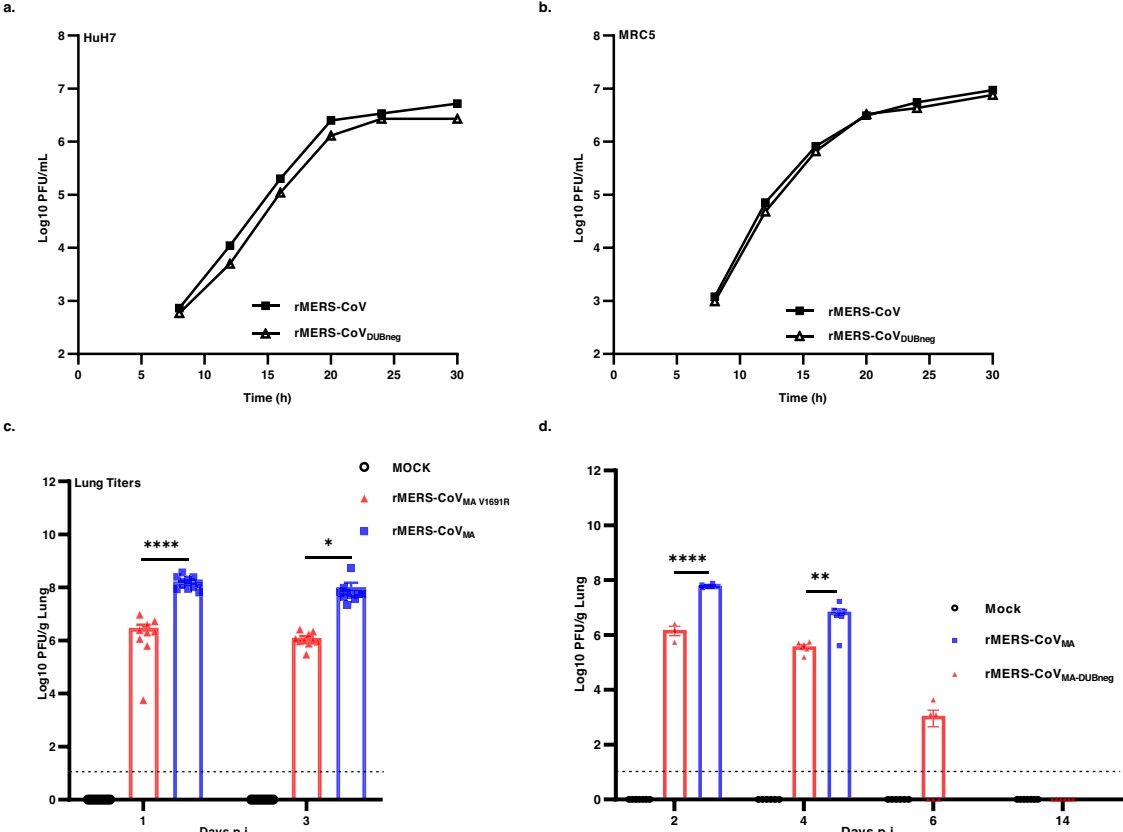

**Fig. 1 | Replication of MERS-CoV DUB-negative virus in vitro and in vivo. a** Huh7 or **b** MRC5 cells were infected with wt (rMERS-CoV) or DUB-negative MERS-CoV (rMERS-CoV$_{DUBneg}$) at a MOI of 5 (Huh7) or 1 (MRC5). At different time points p.i., supernatants were collected and the virus titer was determined by plaque assay on Huh7 cells. Growth curves (**a, b**) were performed twice (*n* = 2 independent replicates) and a representative replicate is shown. **c, d** MERS-CoV DUB-negative infection in hDPP4 KI mice. Groups of 10 (**c**, *n* = 10) or 6 (**d**, *n* = 6) mice were infected intranasally with either rMERS-CoV$_{MA}$ (blue) or DUB-negative rMERS-CoV$_{MA}$ (red) viruses at $10^4$ PFU or mock-infected (black, no virus). Lungs were harvested from mice at the indicated times, then homogenized and virus titers measured by plaque assay on Huh7 cells. The individual virus titers (PFU) per gram of lung tissue and the group means ± SEM are presented at day 1 and 3 (**c**), and days 2, 4, 6, 14 p.i. (**d**). Symbols represent individual mice. The limit of detection for infectious viral progeny is 10 PFU/g Lung and is indicated with a dashed line. An unpaired two-tailed *t* test was used to determine significant differences between the rMERS-CoV$_{MA}$ and the DUB-negative rMERS-CoV$_{MA}$ (day 1 (****$P$ < 0.0001), (day 2(****$P$ < 0.0001), (day 3 (*$P$ = 0.0479), and (day 4 (**$P$ = 0.0092). Source data are provided as a Source Data file.

infection (Fig. 1c, d) indicating that the modified virus is also capable of establishing a robust infection in lungs, albeit less effectively than the parental rMERS-CoV$_{MA}$. Furthermore, over time DUB-negative rMERS-CoV$_{MA}$ was cleared from the lungs. At day 6 p.i. lung virus titers for the modified virus had significantly decreased to ~1×$10^3$ PFU per g of lung tissue for 50% of the animals, while no progeny was measured for the other 50% of the animals at that time point (Fig. 1d). The DUB-negative rMERS-CoV$_{MA}$ infection was completely cleared from the lungs by day 14 (Fig. 1d), while progeny titers for the parental rMERS-CoV$_{MA}$ could not be evaluated after day 4 because all animals had reached the humane endpoint and were euthanized within 4 days post infection. In order to verify whether the DUB-inactivating substitution in the MERS-CoV modified virus was stably maintained in vivo, sequencing of the PLpro-coding region of the viral genome was performed on lung homogenates harvested at day 3 p.i. for both rMERS-CoV$_{MA}$ and DUB-negative rMERS-CoV$_{MA}$ viruses. In both cases, the consensus sequence of the PCR product was found to be identical to the sequence of the BAC-based cDNA clone from which the rMERS-CoV$_{MA}$ or the DUB-negative rMERS-CoV$_{MA}$ had been launched and the V1691R substitution in PLpro had been retained (Supplementary Fig. 2B). Taken together, our data established that the DUB-negative rMERS-CoV$_{MA}$, which stably maintained the V1691R substitution, was replication-competent in vivo and successfully cleared from the lungs of infected mice by day 14 post infection.

## Disrupting DUB activity attenuates MERS-CoV pathogenesis in hDPP4 KI mice

Even though the DUB-negative virus and the wt mouse-adapted parental virus both showed robust replication in the lungs of infected hDPP4 KI mice, the DUB-negative rMERS-CoV$_{MA}$ data suggested that the modified virus was less efficient than its wt counterpart (Fig. 1c, d). This prompted us to evaluate the lethality and disease outcome of the DUB-negative rMERS-CoV$_{MA}$ in the hDPP4 KI model, in which an inoculum of $10^4$ PFU of wt MA virus was previously shown to be lethal in both sexes[36]. We confirmed the lethality of rMERS-CoV$_{MA}$ in both female and male hDPP4 KI mice, aged 8–10 weeks, in a study using intranasal administration of doses ranging from $10^4$ to $10^5$ PFU per animal. All the wt dosages verified in the hDPP4 KI mice had the same disease outcomes: the animals suffered significant weight loss and most died or reached their humane endpoints (Supplementary Fig. 3A, B). To investigate the lethality of the DUB-negative rMERS-CoV$_{MA}$, we infected groups (*n* = 10) of hDPP4 KI mice with a lethal dose of $10^4$ PFU of rMERS-CoV$_{MA}$ or with $10^4$ PFU of DUB-negative rMERS-CoV$_{MA}$ virus, or mock-infected the animals with DMEM only. Animals were then monitored daily over a period of 14 days for clinical signs, body weight loss, and survival. In the 14 days monitoring period, nine out of ten animals infected with rMERS-CoV$_{MA}$ significantly lost weight starting from day 3 p.i. onwards and reached the humane endpoint before day 5 p.i. (Fig. 2a, b). In contrast, all animals infected with the DUB-negative rMERS-CoV$_{MA}$ virus survived and, like the mock-infected animals,

showed no signs of morbidity and kept a relatively stable body weight until the end of the experiment (Fig. 2a, b). It is important to note that the single animal that was infected with the lethal dose of rMERS-CoV$_{MA}$ but unexpectedly survived also did not lose weight nor showed any morbidity (Fig. 2a, b), possibly due to failed or insufficient inoculation or another technical problem.

Next, lungs from mice infected with rMERS-CoV$_{MA}$ or DUB-negative rMERS-CoV$_{MA}$ virus, or mock-infected controls were examined microscopically on day 4 p.i. to investigate virus-induced pathology (Fig. 3 and Supplementary Table S2). In contrast to mock-infected control animals, which exhibited alveolar macrophages only occasionally (Fig. 3a, left panel), the rMERS-CoV$_{MA}$-infected animals typically developed a multifocal to diffuse, predominantly histiocytic, interstitial pneumonia with moderate amounts of macrophages and fewer lymphocytes/plasma cells and viable neutrophils (Fig. 3a, middle panel). The perivascular space of medium-to large-sized blood vessels was markedly expanded by edema in most animals and mild to moderate infiltrates of predominantly lymphocytes/plasma cells and macrophages had formed as perivascular cuffs. Alveolar septa were about twice the size of mock-infected controls and alveolar spaces were filled with increased amounts of macrophages and neutrophils, and in some cases admixed with extravasated erythrocytes (hemorrhage) due to mild septal necrosis. Severe, intra-alveolar, eosinophilic proteinaceous fluid was observed in one animal (Supplementary Table S2, animal 6), as well as hyaline membranes due to fibrin deposition on the alveolar epithelium. Interestingly, the lung response pattern of DUB-negative rMERS-CoV$_{MA}$-infected animals was similar in character to that in rMERS-CoV$_{MA}$-infected animals, albeit less severe and with less interstitial pneumonia-related lesions, which were also more multifocally distributed (Fig. 3a, right panel and Supplementary Table S2). To compare treatment groups, a semi-quantitative combined lung pathology score was calculated, based on the individually scored pneumonia-related lesions for each animal. Using this approach, the group mean for DUB-negative rMERS-CoV$_{MA}$-infected animals was significantly lower than for rMERS-CoV$_{MA}$-infected animals (Fig. 3b).

Taken together, these results indicated that disruption of the DUB activity by the PLpro V1691R substitution in rMERS-CoV$_{MA}$ conferred full survival to infected animals and significantly reduced MERS-CoV-related lung pathology, demonstrating that the DUB-negative rMERS-CoV$_{MA}$ virus was attenuated in vivo.

## Immune responses to infection with DUB-negative rMERS-CoV$_{MA}$

Augmented immune responses have been shown in patients with MERS-CoV[37–39], infected animal models[35,40–43], and cultured cells[44–47] indicating that a delayed IFN signaling and impairment in cytokine and chemokine production may contribute to disease severity. We hypothesized that the complete survival of mice infected with the DUB-negative rMERS-CoV$_{MA}$ and the clearance of the modified virus from infected hDPP4 KI mice were in part the result of enhanced innate immune responses due to the loss of PLpro DUB activity. To examine if infection with the DUB-negative rMERS-CoV$_{MA}$ indeed induced an augmented innate immune response that differed from that elicited by the wt rMERS-CoV$_{MA}$, we evaluated the expression of selected interferon and cytokine proteins in lung tissue homogenates of virus-infected hDPP4 KI mice at 1 and 3 days p.i., samples previously used to determine lung virus titers (Fig. 1c, d). The selected interferons and cytokines were representative for innate immune responses that are relevant for MERS-CoV infection[35,48]. At day 1 p.i., the protein levels of IFNs (IFN-β, IFN-γ and IFNλ) and cytokines (IL-6 and TNF-α) in DUB-negative rMERS-CoV$_{MA}$-infected mice were significantly higher than in rMERS-CoV$_{MA}$-infected mice (Fig. 4). The rMERS-CoV$_{MA}$-infected mice showed a significantly greater increase in protein levels of IFNs (IFN-β and IFNλ) and cytokines (IL-6 and TNF-α) at

day 3 p.i. compared to animals infected with the DUB-negative rMERS-CoV$_{MA}$, which showed drastically reduced levels of IFNs and cytokines that were similar to those in mock-infected mice (Fig. 4). Of note, the DUB-negative rMERS-CoV$_{MA}$ did not induce elevated IL-1β levels at the evaluated days p.i., in contrast to the rMERS-CoV$_{MA}$, which elicited significantly higher IL-1β levels at day 1 p.i. and amounts similar to those induced by DUB-negative rMERS-CoV$_{MA}$ or mock observed at day 3 p.i. (Fig. 4). Furthermore, the lung virus titers for the mice infected with the DUB-negative rMERS-CoV$_{MA}$ were similar at day 1 and 3 p.i. (Fig. 1c), strongly suggesting that the immune responses observed are not a consequence of differences in virus replication in vivo. Interestingly, even though the virus titers in lungs infected with rMERS-CoV$_{MA}$ were significantly higher than for DUB-negative rMERS-CoV$_{MA}$ at day 1 and 3 p.i. (Fig. 1c), the DUB-negative rMERS-CoV rMERS-CoV$_{MA}$ still provoked earlier and increased protein levels of some of the measured IFNs and cytokines (Fig. 4). These results suggest that DUB-negative rMERS-CoV$_{MA}$ infection leads to an altered immune response that generally induces an earlier, and subsequently again downregulated, innate immune response, whereas wt virus-infected mice show a relatively delayed response.

## A single intranasal dose of DUB-negative rMERS-CoV$_{MA}$ elicits potent and sustained neutralizing antibodies

To further evaluate the immune responses to infection with the DUB-negative rMERS-CoV$_{MA}$, the humoral immune response in hDPP4 KI mice was examined. Groups of 8–10-week-old hDPP4 KI mice were bled at day 0 (pre-sera) and intranasally immunized with either $10^4$ PFU of DUB-negative rMERS-CoV$_{MA}$ or rMERS-CoV$_{MA}$, or were mock-immunized with DMEM only as a negative control. Sequential serum samples were collected at weeks 2, 4, 6, and 7 from some animals, and after 7 weeks these were challenged with rMERS-CoV$_{MA}$ while the rest of the animals were further bled at weeks 9 and 11 post immunization (Supplementary Fig. 4). The neutralizing capacity of the induced antibodies was determined using a microneutralization assay against live rMERS-CoV$_{MA}$ as described in "Methods" section. The titer is indicated as the dilution at which an inhibiting effect was visible and the cytopathic effect was fully prevented. The overall immunization scheme is shown in Fig. 5a. All mice immunized with DUB-negative rMERS-CoV$_{MA}$ elicited neutralizing antibodies to rMERS-CoV$_{MA}$ as early as 2 weeks post vaccination and the neutralization titers continued to increase and persist up to 11 weeks post immunization (Supplementary Fig. 4), reaching mean titers of 320 at 7 weeks post immunization (Fig. 5b). In addition, immune sera collected 6 weeks after immunization from the DUB-negative rMERS-CoV$_{MA}$-vaccinated animals, and pooled, also showed potent neutralizing activities against 2 naturally occurring MERS-CoV isolates (EMC/2012 and Jordan N3/2012) and the recombinant wt MERS-CoV derived from EMC/2012 (Fig. 5c). Together, these results indicated that a single intranasal immunization with the DUB-negative rMERS-CoV$_{MA}$ induced potent and sustained neutralizing antibodies against the mouse-adapted rMERS-CoV that were also capable of neutralizing naturally occurring MERS-CoV isolates.

## DUB-negative rMERS-CoV$_{MA}$ vaccination fully protects hDPP4 KI mice against a lethal MERS-CoV challenge with sterilizing immunity

Thus far, we have shown that the DUB-negative rMERS-CoV$_{MA}$ is attenuated and induces robust and persistent neutralizing antibodies against MERS-CoV. We next assessed the protective efficacy of the DUB-negative rMERS-CoV$_{MA}$ as a live-attenuated virus vaccine candidate.

Eight- to ten-week-old hDPP4 KI mice of both sexes were intranasally immunized with $10^4$ PFU of the attenuated virus or mock-vaccinated with DMEM only. Seven weeks after vaccination, mice were challenged with a lethal dose of $10^4$ PFU of rMERS-CoV$_{MA}$ or mock-challenged with DMEM only and monitored daily for weight loss,

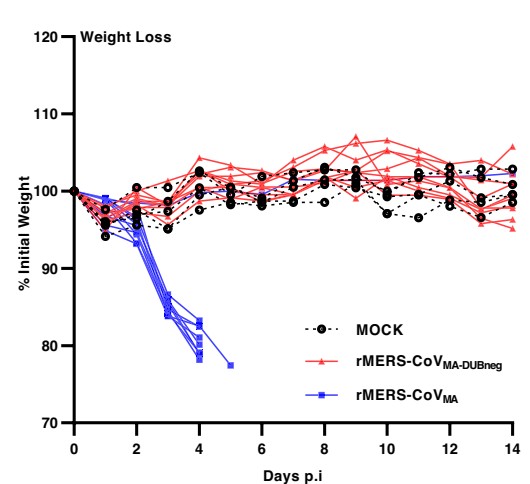

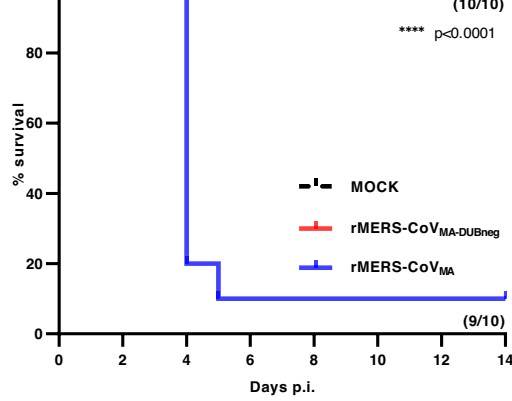

**Fig. 2 | rMERS-CoV$_{MA}$ DUB-negative virus causes attenuated disease in hDPP4 KI mice.** Groups of 10 or 5 mice were infected intranasally with either rMERS-CoV$_{MA}$ (blue) ($n = 10$) or DUB-negative rMERS-CoV$_{MA}$ (red) ($n = 10$) virus at 10⁴ PFU or DMEM (Mock−black) ($n = 5$). **a** Weight loss kinetics after virus challenge and **b** survival percentages were monitored for 14 days. For survival (**b**), a Kaplan–Meier survival plot was made and analyzed with a log-rank test. (****$P < 0.0001$). Source data are provided as a Source Data file.

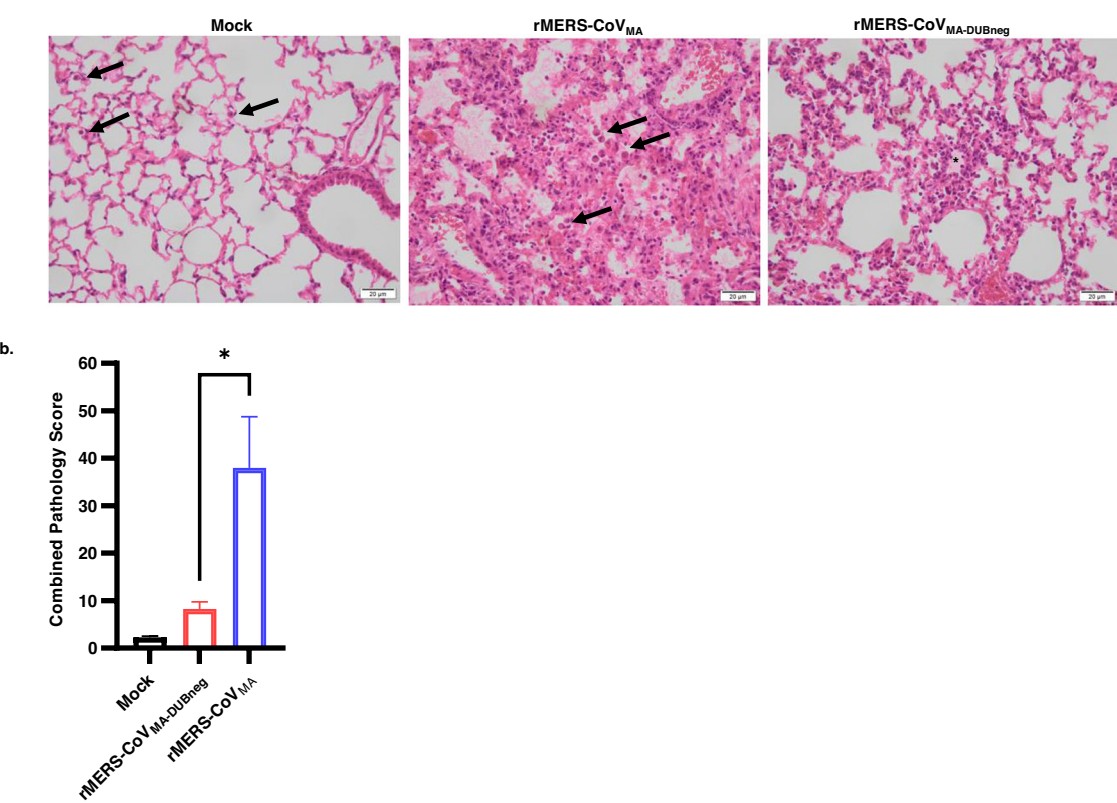

**Fig. 3 | Lung pathology of mock-, rMERS-CoV$_{MA}$-, and DUB-negative rMERS-CoV$_{MA}$-infected hDPP4 KI mice on day 4 post infection.** Groups of 4 animals ($n = 4$) were intranasally inoculated with 50 μL DMEM containing either 10⁴ PFU virus or no virus (mock), sacrificed at day 4 post infection and subsequently examined microscopically. **a** Photomicrographs of representative lung lesions are shown from mock- (left panel), rMERS-CoV$_{MA}$- (middle panel), or rMERS-CoV$_{MA\text{-}DUBneg}$- (right panel) infected animals (H&E stain, ×400 magnification). Compared with the mock-infected animals, which only exhibited occasional alveolar macrophages (arrows), the rMERS-CoV$_{MA}$-infected animals had a diffuse, lymphohistiocytic, interstitial pneumonia with moderate amounts of predominantly macrophages (arrows) and fewer lymphocytes/plasma cells and viable neutrophils. The rMERS-CoV$_{MA\text{-}DUBneg}$-infected animals also had an interstitial pneumonia of similar character, but less extensive in severity and distribution. The asterisk indicates an inflammatory focus with increased numbers of alveolar and interstitial macrophages and viable neutrophils. **b** Bar chart with semi-quantitative combined lung pathology scores. Bar heights indicate group means ($n = 4$) and error bars the standard error of the mean. The difference between the rMERS-CoV$_{MA}$ and rMERS-CoV$_{MA\text{-}DUBneg}$ groups reached statistical significance (Student's $t$ test, unpaired, *$P < 0.0336$). Source data are provided as a Source Data file.

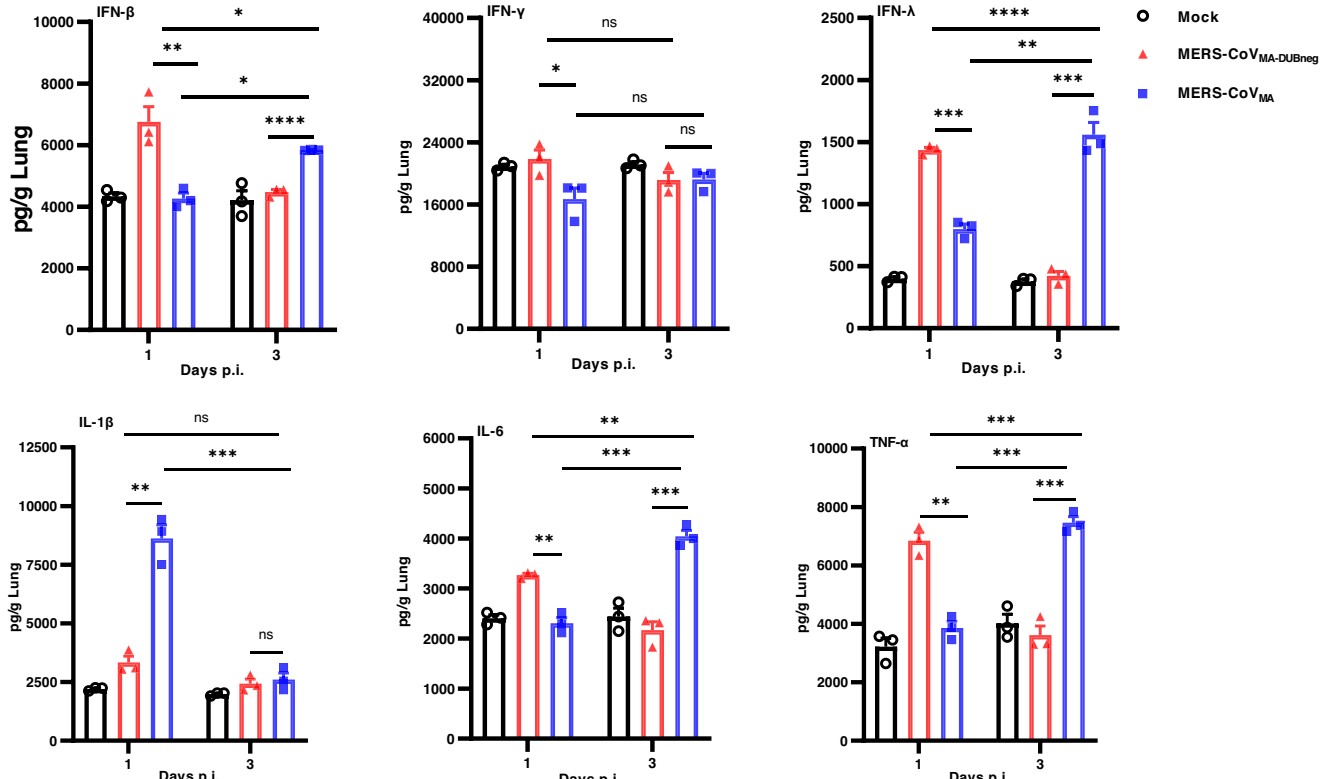

**Fig. 4 | Interferon and cytokine responses to rMERS-CoV_MA or rMERS-CoV_MA-DUBneg infection in hDPP4 KI mice.** The same clarified lung homogenates used for measuring viral titers (see Fig. 1), n = 3, were used to measure levels of selected interferons (IFN-β, IFN-γ, and IFN-λ) and cytokines (IL-1β, IL-6, and TNF-α) by ELISA. Results are presented as means (pg/g lung) ± SEM and compared by one-way ANOVA with Tukey's post hoc test comparisons. (*P < 0.05, **P < 0.01, ***P < 0.001, ****P < 0.0001 and NS: not significant). Source data are provided as a Source Data file.

survival and viral replication (Fig. 6a). While the challenge of mock-vaccinated mice with rMERS-CoV_MA resulted in rapid and significant weight loss by day 4 post challenge, no weight loss or ill-health was observed in DUB-negative rMERS-CoV_MA -vaccinated mice, like in the mock-vaccinated and mock-challenged controls (Fig. 6b). Furthermore, all animals that were vaccinated with the DUB-negative rMERS-CoV_MA and challenged with a lethal dose of rMERS-CoV_MA survived until the end of the experiment (day 14 p.i.), like the mock-vaccinated and mock-challenged controls. In contrast, mock-immunized mice that were challenged with a lethal dose of rMERS-CoV_MA succumbed to infection before day 6 (Fig. 6c). The viral loads in the lungs of DUB-negative rMERS-CoV_MA vaccinated mice at day 2, 4, 6, and 14 after the challenge were below the lower limit of detection (10 PFU/mL), while the mock-immunized and rMERS-CoV_MA-challenged animals had viral loads in their lungs that on average were as high as $3.1 \times 10^8$ PFU at day 2 and $4.1 \times 10^7$ PFU/g at day 4 (Fig. 6d). High levels of both genomic and subgenomic viral mRNA were also detected in the lungs of mock-immunized mice, consistent with the high levels of infectious virus units observed at day 2 and 4 post challenge. In contrast, DUB-negative rMERS-CoV_MA-vaccinated mice had significantly reduced viral RNA levels (at least 5-log reduction) in their lungs at day 2 and 4 post challenge, which were similar to viral RNA levels at days 0, 6, and 14 (Supplementary Fig. 7).

In addition, we investigated lung pathology semi-quantitatively in mock- and rMERS-CoV_MA-DUBneg-vaccinated animals on day 4 following lethal challenge with $10^4$ PFUs of rMERS-CoV_MA. Mock-vaccinated animals developed a moderate to severe, diffuse, predominantly histiocytic interstitial pneumonia with some variability in severity and extent of scored lesions between animals (Supplementary Fig 5 and Supplementary Table S3). This aligned with the lung pathology observed in the previous experiment (Fig. 3 and Supplementary

Table S2) and was expected. In contrast, the rMERS-CoV_MA-DUBneg-vaccinated animals developed less interstitial pneumonia-related lesions, which were generally also less in severity and distribution compared with the mock-vaccinated animals (Supplementary Fig 5A and Supplementary Table S3). Overall, the combined pathology scores of rMERS-CoV_MA-DUBneg-vaccinated animals were generally lower than for mock-vaccinated animals (Supplementary Fig 5B) and the difference between the means of both groups was statistically highly significant (P<0.001, unpaired, two-tailed Student's t test). Taken together, these results showed that a single intranasal immunization with live-attenuated DUB-negative rMERS-CoV_MA resulted in limited lung pathology and complete protection against a lethal dose of rMERS-CoV_MA with sterilizing immunity.

**Passive transfer of sera from mice immunized with the DUB-negative rMERS-CoV_MA protects naive hDPP4 KI mice from a lethal MERS-CoV infection**

To evaluate the protective efficacy of antibodies induced by immunization with DUB-negative rMERS-CoV_MA, passive transfer studies were performed in naive hDPP4 KI mice (Fig. 7a). To this end, mouse sera collected at 4 weeks after immunization with DUB-negative rMERS-CoV_MA (Figs. 5b and 7b) were pooled and diluted 1:10. Eight- to ten-week-old naive hDPP4 KI mice were then injected intraperitoneally with 250 μL of the pooled mouse sera (indicated as "treated") and control groups (indicated as "mock-treated") received the same volume of pooled serum collected from mock-vaccinated animals. On day 1 after treatment, mice were intranasally challenged with a lethal dose of $10^4$ PFU of rMERS-CoV_MA and monitored for weight loss, survival and viral replication in the lungs. Both treated and mock-treated groups of animals initially lost weight, however, the mock-treated mice lost significantly more weight, and all succumbed to disease by 4 days

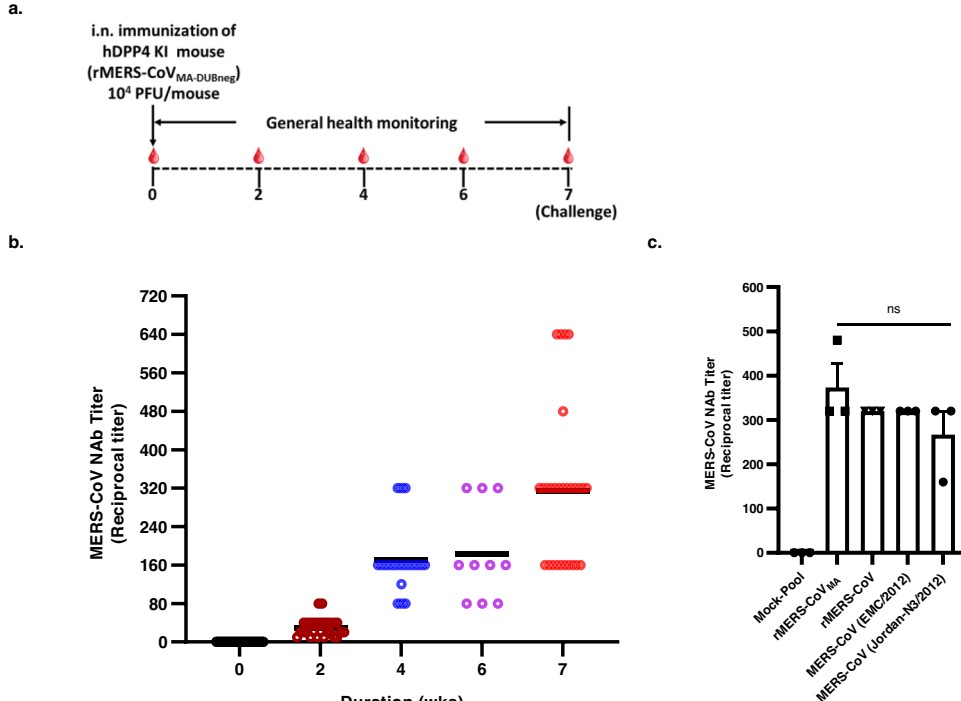

**Fig. 5 | Neutralizing antibody responses induced by DUB-negative rMERS-CoV$_{MA}$ vaccination in hDPP4 KI mice. a** Timeline for the immunization. **b** Groups of 28 or 10 mice were immunized intranasally with either 10$^4$ PFU of rMERS-CoV$_{MA-DUBneg}$ or mock-immunized. At week 0, 2, 4, 6, and 7, blood was collected and twofold serially diluted serum samples were analyzed for neutralization activity over time against rMERS-CoV$_{MA}$ in Huh7 cells. Black horizontal lines indicate mean ± SEM reciprocal titers and colored circles indicate individual values per group (week 0, 2, 4, and 7: $n = 28$; week 6: $n = 10$) (**c**). Mean neutralizing antibody titers ($n = 3$), against rMERS-CoV$_{MA}$, rMERS-CoV, MERS-CoV (EMC/2012) and MERS-CoV (Jordan N3/2012), in blood collected at day 28 post immunization. Error bars indicate the SEM. None of the group comparisons reached statistical significance (unpaired two-tailed Student's $t$ test). Source data are provided as a Source Data file.

post challenge while most mice (5/7) that received neutralizing antibody serum rapidly regained weight in the ensuing period and survived until the end of the experiment (Fig. 7c, d). While the viral lung titers on day 2 post challenge were similar for both groups, on day 4 these titers were significantly lower in all animals that had received serum from DUB-negative rMERS-CoV$_{MA}$-immunized mice compared to animals that had received mock serum (Fig. 7e). There were two mice in the group of animals that received neutralizing serum that lost weight rapidly following viral challenge and were euthanized before reaching the end of the experiment (Fig. 7c). The weight loss and survival kinetics of these two animals that received neutralizing serum (treated) mirrored the kinetics of the untreated group (Fig. 7c, d), which suggested that passive transfer of intraperitoneally injected neutralizing antibodies in these two animals may have been unsuccessful.

Overall, these results confirmed that virus-neutralizing antibodies induced by vaccination with the DUB-negative rMERS-CoV$_{MA}$ play a crucial role in protecting hDPP4 KI mice from a lethal challenge with rMERS-CoV$_{MA}$.

## Discussion

MERS-CoV is highly pathogenic and continues to be sporadically transmitted to humans resulting in severe disease with high fatality[49]. Despite the potential pandemic threat that MERS-CoV continues to pose, there are still no approved vaccines for use in humans. Thus, a strong case can be made for continued efforts to study the pathogenesis of this virus and develop durable and potent MERS-CoV vaccines. Here, we have performed a proof-of-concept study demonstrating that the elimination of the innate immune-evasive DUB activity of MERS-CoV PLpro may offer a strategy to develop a modified live vaccine.

We and others have shown that the PLpro found in all coronaviruses, including MERS-CoV, is highly conserved and multifunctional. It is not only capable of cleaving the viral replicase polyproteins, but it is also a deubiquitinase (DUB) that removes Ub-like modifiers such as ISG15, presumably to suppress innate immune responses[50–52]. In our previous work, based on the crystal structure of the PLpro–Ub complex, we were able to design mutations that allowed us to functionally separate the DUB activity of MERS-CoV PLpro from its replicase polyprotein cleavage activity. The fact that the DUB-negative rMERS-CoV PLpro was less able to downregulate IFN-β promoter activity was consistent with involvement of its DUB activity in suppressing innate immune responses[32]. We therefore hypothesized that the PLpro DUB activity functions as a coronaviral interferon antagonist, and that the disruption of this activity might attenuate MERS-CoV and serve as the basis to generate a new class of coronavirus MLV vaccines. To that end, we engineered a recombinant DUB-negative MERS-CoV and analyzed the replication of this modified virus in cell culture and evaluated its pathogenicity in the hDPP4 KI mouse model. Our data shows that the DUB-negative rMERS-CoV$_{MA}$ replicates to comparable levels as wt virus in cell culture, but less effectively in mouse lungs (Fig. 1). Interestingly, the protein levels for IFN-β, IFN-λ, and some pro-inflammatory cytokines (IL-6 and TNFα) were significantly higher in the lungs of DUB-negative rMERS-CoV$_{MA}$ infected mice at 1 day p.i. compared to wt virus-infected mice (Fig. 4). In DUB-negative rMERS-CoV$_{MA}$-infected mice, the protein levels of these IFNs and pro-inflammatory cytokines decreased to similar levels as those obtained from mock-infected mice at 3 days p.i. However, in the wt virus-infected mice these levels were significantly increased (Fig. 4). This data suggests that DUB-negative rMERS-CoV$_{MA}$ infection leads to an altered innate immune response that induces an earlier and subsequently downregulated response with diminished lung pathology

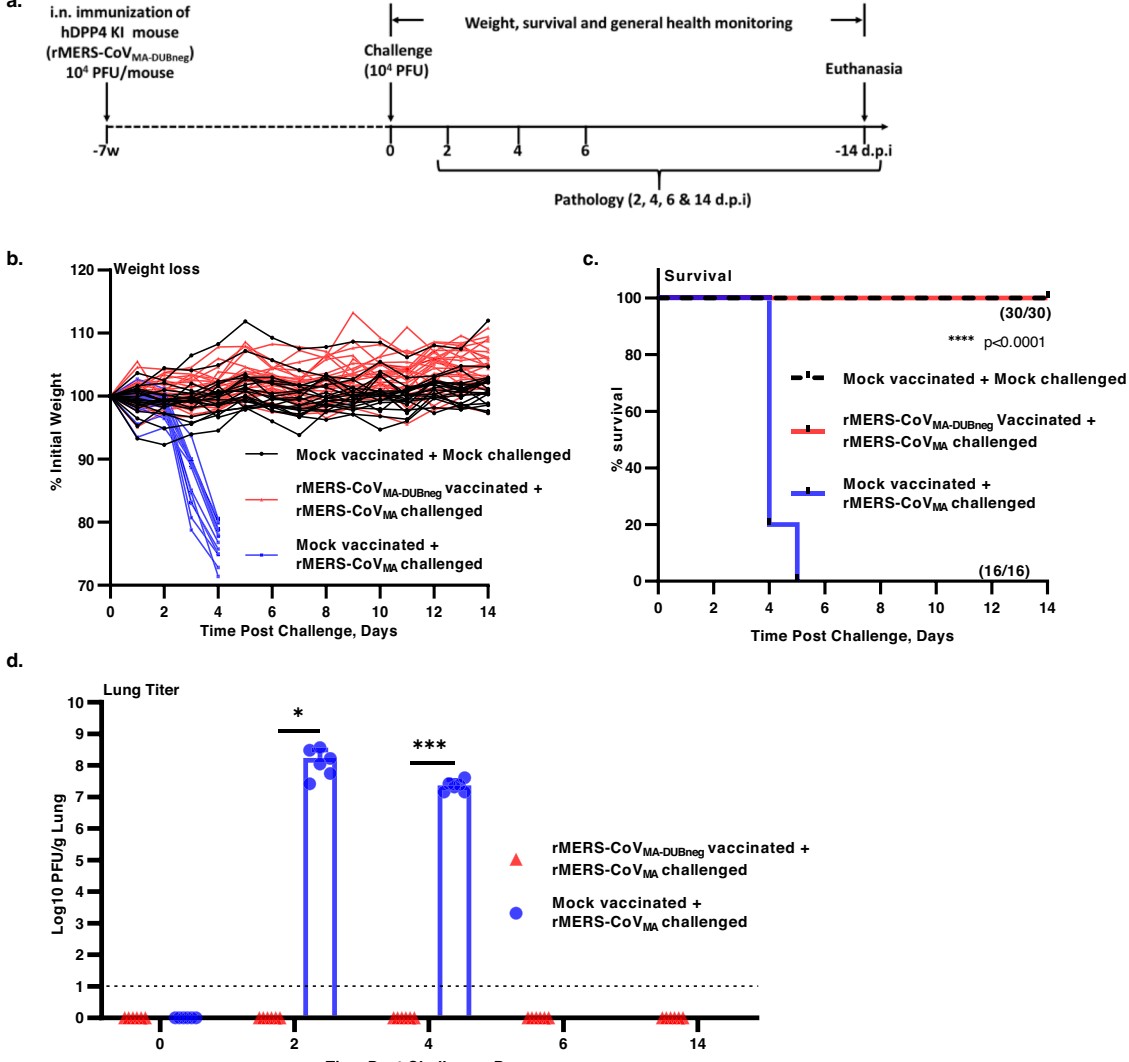

**Fig. 6 | Challenge of hDPP4 KI mice after a single vaccination with the DUB-negative rMERS-CoV$_{MA}$. a** Timeline for immunization, challenge and evaluation of protective efficacy. At 49 days after immunization, mice were intranasally challenged with a lethal dose $10^4$ PFU of rMERS-CoV$_{MA}$ or mock-challenged with DMEM and monitored daily for (**b**) weight loss (% from initial weight), (mock-vaccinated and mock-challenged: $n = 21$; mock-vaccinated and rMERS-CoV$_{MA}$ challenged: $n = 16$; rMERS-CoV$_{MA-DUBneg}$ vaccinated and rMERS-CoV$_{MA}$: $n = 30$), **c** survival (%) and disease symptoms for 14 days. Statistical comparisons between means were performed by a log-rank test and corrected for multiple comparisons. ****$P < 0.0001$. **d** On days 0, 2, 4, 6, and 14 post challenge, lung tissues were collected for virus

titration by plaque assay on Huh7 cells. The individual virus titers (PFU) per gram of lung tissue and the mean per group $\pm$ SEM are presented for day 0, 2, 4, 6, and 14 p.i. (**d**). Symbols represent individual mice ($n = 6$). The limit of detection for infectious viral progeny is 10 PFU/g Lung and is indicated with a dashed line. An unpaired two-tailed $t$ test was used to determine significant differences between the mock-vaccinated and rMERS-CoV$_{MA}$-challenged (shown in blue) and the DUB-negative rMERS-CoV$_{MA}$-vaccinated rMERS-CoV$_{MA}$ challenged (shown in red). Day 2 (*$P < 0.0117$) and day 4 (***$P < 0.0002$). Source data are provided as a Source Data file.

(Fig. 3) and clinical disease (Fig. 2), whereas wt virus-infected mice showed a relatively delayed and still robust response with higher viral loads and severe disease (Figs. 1–3). An augmented innate immune response during the early stages of SARS-CoV and MERS-CoV infection has been reported[48], with the type I IFN response to viral infection being suppressed, as these viruses employ multiple strategies to interfere with the signaling that leads to type I IFN production[53]. This dampening strategy is speculated to be closely associated with viral dissemination from the upper to the lower respiratory tract and disease severity[48]. Further studies in appropriate animal models will have to establish whether the DUB-negative virus replicates to similar levels as wt virus in the nasal turbinates and upper respiratory tract. This would allow for the evaluation of the role of the DUB activity in early innate immune responses and related effects on viral dissemination, revealing perhaps why viral loads of the modified virus

were lower compared to wt virus in the lung. It is also likely that the V1691R mutation in PLpro not only impairs the DUB activity, but also affects the deISGylating (deISG) activity in vivo, contributing to an altered innate immune response (Fig. 4) and attenuation of MERS-CoV virulence (Fig. 2)[54,55]. Other cellular processes besides innate immunity may also be influenced by PLpro's DUB activity, which might further contribute to its attenuation in vivo. Therefore, it remains important to identify and characterize cellular protein targets that are deubiquitinated by PLpro. Several studies have shown that ubiquitination plays a crucial role in the regulation of both the activation and the attenuation of innate immune responses to viral infection[23], but so far the DUB activity in the context of infection has been examined for only a few viruses, mainly those encoding DUBs that belong to the ovarian tumor domain-containing family (OTUs), including equine arteritis virus (EAV)[56,57] and Crimean-Congo

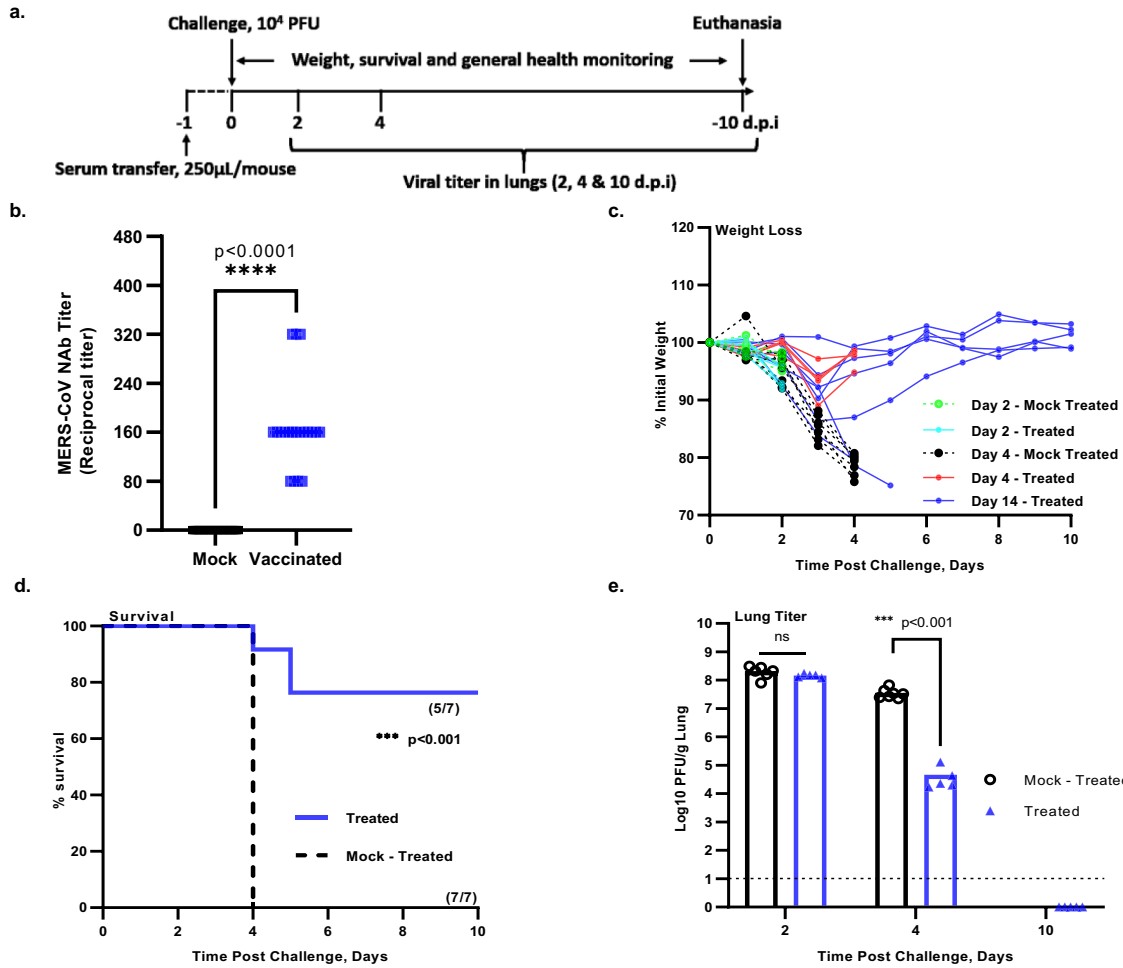

**Fig. 7 | Challenge of hDPP4 KI mice after passive transfer of immune sera from mice immunized with the DUB-negative rMERS-CoV$_{MA}$. a** Timeline of immunization, serum transfer, challenge and clinical outcomes. Naive hDPP4 KI mice were injected intraperitoneally with 0.25 mL of pooled serum from hDPP4 KI mice immunized with a single shot of the rMERS-CoV$_{MA-DUBneg}$. At 24 h post serum transfer, mice were intranasally challenged with a lethal dose of $10^4$ PFU of rMERS-CoV$_{MA}$. **b** Neutralizing activity of the pooled immune sera from hDPP4 KI mice immunized with rMERS-CoV$_{MA-DUBneg}$ (pooled mock-vaccinated: $n = 28$; pooled rMERS-CoV$_{MA-DUBneg}$: $n = 20$). Black horizontal lines indicate mean reciprocal titers and squares (symbols) indicate individual values per group. An unpaired two-tailed $t$ test was used to determine significant differences between the pooled mock-vaccinated neutralizing titers (black squares) and the pooled DUB-negative rMERS-CoV$_{MA}$ vaccinated neutralizing titers (shown in blue squares), ****$P < 0.0001$. **c** Body weight kinetics after virus challenge and **d** survival ($n = 7$) were monitored daily for 10 days. **e** On days 2, 4, and 10 post challenge, lung tissues were collected for virus titration by plaque assay on Huh7 cells. The mean ± SEM per group and the virus titer in PFU per gram of lung tissue are presented. Symbols represent individual mice. The limit of detection for infectious viral progeny is 10 PFU/g lung and is indicated with a dashed line. Statistical comparisons between means were performed by Student's $t$ test (unpaired tqo-tailed): ***$P < 0.001$. Source data are provided as a Source Data file.

hemorrhagic fever virus (CCHFV)[58]. The DUBs of the OTU family are structurally different from CoV PLpros, which belong to the ubiquitin-specific protease DUB family. More recently, the DUB-mediated innate immune evasion activities of SARS-CoV PLpro[59] and MHV PLP2[60] were investigated. The structural equivalent of the DUB-negative rMERS-CoV$_{MA}$ investigated here (i.e., the V1691R substitution) was introduced into SARS-CoV PLpro, yielding a modified virus with a M1748R substitution (using SARS-CoV pp1a/pp1ab amino acid numbering). Compared to the wt control, replication of this modified virus was slightly delayed in cell lines that can mount an innate immune response, and it induced higher IFN-β mRNA levels[59]. As in our previous studies[32], using structure-guided mutagenesis, the DUB activity of MHV PLP2 was found to negatively regulate the IFN response in macrophages, however, the DUB-negative MHV was only mildly attenuated in a mouse model[60]. This suggests that different PLpro/DUB domains have unique antagonistic effects on the innate immune response, depending on the virus and host cell type. Intriguingly, using the hDPP4 KI mouse model, we showed that all mice infected with the MERS-CoV DUB-negative rMERS-CoV$_{MA}$ survived while all mice infected with the wt virus succumbed to the infection (Fig. 2). The interstitial pneumonia-related lesions observed in the lungs of mice infected with the DUB-negative rMERS-CoV$_{MA}$ were significantly less in severity and distribution when compared with wt virus-infected controls, and this was in line with the better survival rates of mice infected with the DUB negative virus relative to wt virus-infected mice (Fig. 3). Therefore, it is likely that the survival rate differences are due to an accelerated, and after induction properly downregulated, innate immune response in DUB-negative rMERS-CoV$_{MA}$-infected mice. This response likely is more effectively clearing the virus, impairing viral dissemination and progression of disease along the airways into the lungs, and thereby limiting tissue damage and thus protecting these mice from lethality (Figs. 1–4). These results clearly suggest that the DUB activity of MERS-CoV PLpro is a prominent virulence factor, and that its disruption leads to a live-attenuated virus that can serve as the basis to generate a new class of modified live vaccines for coronaviruses.

Live-attenuated vaccines generally have the advantage of requiring only a single dose, which induces rapid and long-lasting immunity against a viral disease. Unlike the vaccines targeting SARS-CoV-2, the majority of MERS-CoV vaccine candidates are still in preclinical stages and only a few, including the GLS-5300, ChAdOx1 MERS, and MVA-MERS-S, have undergone Phase 1/2 trials[10,61]. Many of these vaccine candidates are virus vectored (e.g., ChAdOx1 MERS and MVA-MERS-S) or DNA based (e.g., GLS-5300) and almost all of these candidates are Spike protein-based MERS-CoV vaccines[61]. The long-term protection offered by these subunit-based vaccines has not been fully established, but in general it is expected to be not long-lasting. This has been illustrated by the limited ability of the available licensed COVID-19 vaccines to protect against infection and symptomatic illness over time necessitating the use of boosters and fueling worldwide vaccination disparities. The ongoing evolution of SARS-CoV-2 and the pandemic threat posed by other circulating coronaviruses like MERS-CoV call for the development of more effective vaccines. Modified live-attenuated vaccines are multivalent, induce strong humoral and cell-mediated immune responses, which contribute to long-lasting immunity while the risk of reversion to virulence can be minimized[62]. Recently, a single dose of live-attenuated MERS-CoV carrying partial deletions in the E protein provided full protection against a lethal dose of MERS-CoV in hDPP4 KI mice[63]. The immune responses induced by this live-attenuated MERS-CoV still need to be determined and the durability of all the candidate MERS-CoV vaccine-induced immune responses and precise correlates of protection also remain to be defined. Our results demonstrate that a single intranasal immunization with the DUB-negative rMERS-CoV$_{MA}$ in mice is sufficient to induce robust and persistent neutralizing antibody responses (suggestive of sustained germinal center reactions), which last up to at least 11 weeks (Fig. 5). The neutralizing antibody responses were seemingly increased at week 7 (Fig. 5) while the animals were not boosted, possible due to the use of a lower infective dose of 55 TCID$_{50}$ at week 7 (measured by backtitration of the virus dose), compared to the targeted dose of 120 TCID$_{50}$ MERS-CoV. The persistence of effective levels of neutralizing antibodies after a single dose is superior to other candidate vaccines for which two or multiple rounds of immunization are required to induce detectable neutralizing antibodies that similarly confer complete protection against MERS-CoV challenge[64]. The DUB-negative rMERS-CoV$_{MA}$ induced protective and sterilizing immunity against a lethal MERS-CoV challenge (Fig. 6), which is further supported by the lack of clinical symptoms and weight loss, and the limited lung pathology observed in a stringent and lethal mouse model. MERS-CoV transmission largely occurs through the respiratory route[65] and intranasal administration of the DUB-negative rMERS-CoV$_{MA}$ therefore likely generated adequate mucosal immunity to neutralize the virus at the site of inoculation, thus conferring complete protection against infection. Future studies and other animal models of MERS-CoV will have to establish whether the DUB-negative rMERS-CoV$_{MA}$ also can prevent transmission by measuring both infection and innate immune responses in the upper respiratory tract after challenge. In an effort to establish the correlates of protection for the DUB-negative rMERS-CoV$_{MA}$, our data demonstrate the seminal role of protective antibodies, as passive transfer of immune sera from DUB-negative rMERS-CoV$_{MA}$-immunized mice to naive hDPP4 KI mice decreased viral loads in the lungs and also provided a substantial survival advantage against a lethal MERS-CoV challenge (Fig. 6). Passive transfer of immune serum from these mice protected up to 71% of naive mice from lethal MERS-CoV infection, suggesting that T-cell responses or high neutralizing antibody titers are required for sufficient protection. This is also corroborated by limited studies suggesting that recovery from MERS-CoV infection is associated with both antibody and T-cell responses and that only high titers of neutralizing activity can suppress viral replication and subsequent spread[66]. Nonetheless, while our data clearly show that neutralizing antibody responses play a key part in protection, a single dose of the DUB-negative rMERS-CoV$_{MA}$-also induced IFN-γ producing T cells as measured by ELISpot (Supplementary Fig. 6). It is likely that both neutralizing antibody and cellular immune responses elicited by the DUB-negative rMERS-CoV$_{MA-DUBneg}$ play a crucial role in protection against a lethal MERS-CoV infection in mice. Finally, the V1691R mutation introduced in MERS-CoV PLpro appears to confer reasonable stability against possible reversion of the attenuated DUB-negative phenotype to wt virulence, as assessed by (next-generation) full genome sequencing of serially passaged virus in vitro. Advances in the ability to engineer live-attenuated viruses against reversion to virulence have been described recently for various live-attenuated vaccines[67] and some of these methods could be employed to improve the long-term stability and safety profile of the DUB-negative virus while maintaining live attenuation. These methods may include finding and using a combination of multiple DUB knock-out mutations that independently can provide a full DUB knockout phenotype. Additional mutations can also be introduced in other coronavirus interferon antagonists like the viral nsp16[31], nsp1[68], or nsp15[69], or removal/partial deletion of the E protein gene or accessory protein-encoding ORFs[36]. The introduction of multiple point mutations to enhance the safety profile of the DUB-negative MERS-CoV might be preferred to the complete removal of some ORFs, including ORF5 as a previous mouse-adapted virus that was engineered with a full deletion of ORF5 was more virulent in mice[36]. Furthermore, the transcription regulatory sequences of the DUB-negative MERS-CoV could be engineered to reduce the likelihood of successful recombination with other coronaviruses[70]. Importantly however, considering that each of these mutations may further attenuate the virus, one will have to find the right balance between adding additional sequence modifications for safety on the one hand and running the risk of over-attenuation on the other.

Overall, our data show for the first time that the removal of the DUB activity, an interferon antagonist of MERS-CoV, leads to a live-attenuated MERS-CoV with promising features. The DUB-negative rMERS-CoV$_{MA}$ induces potent neutralizing antibodies, can be administered intranasally with a single dose, and can protect against severe MERS-CoV infection and lung disease with sterilizing immunity in a lethal mouse model, supporting its further development as a vaccine. In light of the progressed SARS-CoV-2 pandemic, the global build-up of human immunity against this virus and its variants, and the limited longevity of protection inferred by current vaccines against re-infection, there is room for new vaccine designs with improved features. These may well include modified live virus vaccines, administered through mucosal or oral routes, which often display superior characteristics as discussed above, and provide opportunities to ensure safety. The MERS-CoV MLV vaccine strategy proposed here may therefore also offer an effective strategy for COVID-19 pandemic management in advanced stages and for other coronaviruses of societal and/or economic importance.

## Methods

### Ethics declaration

All experiments involving animals were approved by the Animal Experiments Committee of the LUMC and performed according to the recommendations and guidelines set by the LUMC, the Dutch Experiments on Animals Act, and were in strict accordance with EU regulations (2010/63/EU).

### Cell culture

Huh7 cells (a kind gift of Dr. Ralf Bartenschlager, Heidelberg University, purchased from JCRB, No. JCRB0403) were grown in Dulbecco's modified Eagle's medium (DMEM; Lonza) supplemented with 8% fetal calf serum (FCS; Bodinco BV), 100 units/ml penicillin (Lonza), 100 units/ml streptomycin (Lonza), 2 mM L-glutamine and non-essential amino acids (both PAA) at 37 °C and 5% CO$_2$. MRC5 cells (ATCC

CCL-171) were cultured in Eagle's minimum essential medium (EMEM; Lonza) with the same supplements as used in the medium for Huh7 cells and grown at 37 °C and 5% $CO_2$. BHK-21 cells (C-13, CCL-10, ATCC) were grown in Glasgow minimum essential medium (Gibco) supplemented with 8% FCS, 100 units/ml penicillin, 100 units/ml streptomycin, 10% tryptose phosphate broth (Gibco), and 10 mM HEPES (pH 7.4; Lonza) at 37 °C and 5% $CO_2$.

## Construction and launch of recombinant-modified MERS-CoV

Recombinant rMERS-CoV and rMERS-CoV$_{DUBneg}$ were derived from MERS-CoV full-length cDNA clones based on MERS-CoV strain EMC/2012[71] and the recombinant mouse-adapted viruses rMERS-CoV$_{MA}$ and rMERS-CoV$_{MA-DUBneg}$ were generated from pBAC-MERS$^{FL}$-MA-5FL full-length cDNA clones[36] based on the mouse-adapted MERS$_{MA}$6.1.2 virus[35]. A full-length MERS-CoV cDNA in a bacterial artificial chromosome (BAC) vector was previously equipped with a T7 RNA polymerase promoter and a unique 3'-terminal NotI site for run-off transcription. After passaging of the MERS-CoV EMC/2012 isolate in Vero cells, a premature stop codon in ORF5 was observed in the minority of sequence reads, but this substitution became fixed upon additional virus passaging[34,35]. The engineering and characterization of the mouse-adapted MERS-CoV infectious cDNA (pBAC-MERS$^{FL}$-MA) was previously described[36]. To avoid complications due to ORF5 evolution[34,35] and associated changes in host immune suppression while generating the PLpro-modified virus stocks for our studies[30,71–73], all PLpro-modified viruses were engineered using a full-length construct containing the premature stop codon at ORF5 codon 108. These clones, pBAC-MERS-CoV-ORF5stop, and pBAC-MERS-CoV$_{MA}$-ORF5-stop, were generated by two-step en-passant in vivo recombineering reactions in *E. coli*[33]. Subsequently, substitutions in the PLpro-coding sequence were introduced in these clones using the same procedure. Bacmids were isolated from bacteria and sequenced to verify the presence of the introduced substitution(s). pBAC-MERS-CoV-ORF5-stop, pBAC-MERS-CoV-MA-ORF5stop and derivatives thereof were linearized with *NotI* and then purified by phenol-chloroform extraction and ethanol precipitation. Approximately 1 µg of DNA was used as a template for in vitro transcription using the mMESSAGE mMACHINE T7 transcription kit (Thermo Fisher Scientific). BHK-21 cells (C-13, CCL-10, ATCC) ($5 \times 10^6$) were electroporated with 5 µg of this in vitro transcribed RNA using the Nucleofector 2b device (Lonza; program T-020) with Cell Line Nucleofector Kit T (Lonza). Immediately after electroporation, the cells were taken up in a prewarmed cell culture medium and mixed with Huh7 cells (JCRB, No. JCRB0403) ($5 \times 10^6$) before seeding in a 75-cm$^2$ flask. Virus-containing supernatants were collected when complete cytopathogenic effect (CPE) was observed, usually 3 to 4 days after electroporation. The harvested virus was passaged once in Huh7 cells (JCRB, No. JCRB0403) to grow a virus stock for further experiments.

## rMERS-CoV titration and sequencing

rMERS-CoV titers were determined by plaque assay on Huh7 (JCRB, No. JCRB0403) or MRC5 cells (CCL-171, ATCC). Huh7 cells were fixed after an incubation period of 3 days, whereas MRC5 cells were fixed after 4 days. The tissue culture infective dose 50 (TCID$_{50}$) endpoint dilution method in Huh7 cells and the TCID50 was calculated by the Spearman–Kärber algorithm. All work with live MERS-CoV was performed in the biosafety level 3 facility at the Leiden University Medical Center (LUMC). In order to confirm the presence of the intended substitutions in the rMERS-CoV PLpro- and rMERS-CoV$_{MA}$ PLpro- coding sequences, RNA was isolated from virus-containing supernatants or lung homogenates with the QIAamp Viral RNA Mini Kit (Qiagen). Total RNA was reverse transcribed to cDNA using RevertAid H minus reverse transcriptase (Thermo Fisher Scientific) and random hexamers. The PLpro domain (nucleotides 4435-5409 of the MERS-CoV genome) was amplified by PCR using Accuzyme DNA

polymerase (Bioline) and after purification the PCR product was sequenced.

## Evaluation of rMERS-CoV replication

Huh7 cells (JCRB, No. JCRB0403) were infected with wt or modified rMERS-CoV at a multiplicity of infection (MOI) of 0.01 or 5, to analyze multi- or single-cycle infections, respectively. MOI 0.01 or 1 was used to infect MRC5 cells (CCL-171, ATCC). The rMERS-CoV inoculum was prepared in PBS containing DEAE (0.005% w/v) and 2% FCS, which was put on the cells after removing the medium. Inocula were removed after 1 h at 37 °C and EMEM supplemented with antibiotics, and 2% FCS was added to the cells. Following a high-MOI infection, cells were first washed three times with PBS before adding medium. Supernatants were harvested at various time points and rMERS-CoV titers were determined by plaque assay on Huh7 cells (JCRB, No. JCRB0403). Significance relative to the wt virus control was calculated using an unpaired Student's *t* test and *P* values of less than 0.05 were considered statistically significant.

## Infection of hDPP4 KI mice with rMERS-CoV$_{MA}$ and DUB-negative rMERS-CoV$_{MA}$

Viruses were grown on Huh7 cells (JCRB, No. JCRB0403) and titrated by plaque assay on Huh7 cells as described above. Human DPP4-KI mice[35] were bred and maintained at the LUMC Central Animal Facility (PDC). All experiments with infected animals were performed in a class 3 biological safety cabinet in the Animal BSL-3 unit of the LUMC Central Animal Facility (DM3 unit). Mice were housed in individually ventilated isolator cages (IsoCage Biocontainment System, Tecniplast) under specified pathogen-free conditions with ad libitum access to food and water and cage enrichment at 20–22 °C, a humidity of 45–65% RV and a light cycle of 6:30 h–7:00 h sunrise, 07:00 h–18:00 h daytime and 18:00 h–18:30 h sunset.

Male and female mice (C57BL/6NTac-Dpp4tm3600(DPP4)Arte) aged 8–12 weeks at the start of the experiment were randomized from different litters into experimental groups, and were acclimated at the BSL-3 facility for 7 days before the experiments. Mice were anesthetized with isoflurane and infected intranasally (i.n.) with $10^4$ PFU/mouse (or another dose as specified) diluted in DMEM to a volume of 50 µL. Control mice were mock-infected with 50 µl DMEM only. Infected mice were monitored daily for weight loss, disease symptoms and survival. Mice that reached 20% weight loss were deemed to have reached their humane endpoints and euthanized. Mice that were considered moribund were euthanized at the discretion of the researcher and designated veterinarian. The endpoint of survival experiments was set at 14 days post inoculation (p.i.), except for mice that died or reached a humane endpoint before that time. At various time points, mice were euthanized by intraperitoneal injection (i.p.) of sodium pentobarbital (Euthasol 200 mg/kg) under isoflurane anesthesia to collect lungs and spleens at necropsy. Lungs were examined for macroscopic lesions and dissected into three parts, one for virus titration, one for immune responses, and one for histopathology.

## Immunization and challenge of hDPP4 KI mice

Eight-to-twelve-week-old mice (C57BL/6NTac-Dpp4tm3600(DPP4) Arte) of both sexes were randomly divided into two groups. Mice in the experimental group were intranasally (i.n.) immunized with DUB-negative rMERS-CoV$_{MA}$ under isoflurane anesthesia ($10^4$ PFU in 50 µL DMEM) and mice in the control group received the same volume of DMEM without virus. Blood sampling for serum isolation and neutralizing antibody determination was collected on weeks 0, 2, 4, 6, 7, 9, and 11 via tail-cut vein under isoflurane anesthesia. At 7 weeks post immunization, DUB-negative rMERS-CoV$_{MA}$- and mock-immunized mice were challenged with a lethal dose of rMERS-CoV$_{MA}$ ($10^4$ PFU in 50 µL DMEM) or mock inoculated with DMEM only as described above. Morbidity/mortality status and weights were assessed and recorded

daily and at days 0, 2, 6, and 14, lungs were collected for virus titration by plaque assay and histopathology.

## Lung virus titers

To determine virus titers, lungs were weighed and placed in a gentleMACS M Tube (Miltenyi Biotec) containing 2 ml of PBS with 100 units/ml penicillin, 100 units/ml streptomycin (Lonza), 50 μg/ml gentamycin (Sigma-Aldrich), and 0.25 μg/ml Fungizone (Gibco). Lung tissues were homogenized with the gentleMACS dissociator by running program Lung_02 (Miltenyi Biotec, Inc.). Supernatants from homogenized samples were pre-clarified at $300 \times g$ for 1 min and then further centrifuged at $10,000 \times g$ for 5 min. Thereafter, infectious virions in the lung were determined by plaque assay on Huh7 cells (JCRB, No. JCRB0403) as described above and expressed as PFU/g lung for MERS-CoV. The quantification of MERS-CoV viral RNA was performed using lung homogenates lysed with TriPure isolation reagent (Roche Applied Science) in gentleMACS M Tubes (Miltenyi Biotec) according to the manufacturer's instructions. The subgenomic mRNA PCR primers (forward- GTACCTCTTAATGCCAATTC- and reverse- GAGCCAGTTGCxTTAATTC) and probe (TexasRed/TCTGTCCTGTCTCCGCCAATAC /BHQ2) targeting the nucleocapsid (N) protein gene and the genomic RNA PCR primers (forward- CCGACTCTCTTTAGACTTA- and reverse- ACAGCATGAATGTTGTAC) and probe (FAM/TAACACTTCTTACAGCAGCAACCTC/BHQ1) targeting ORF1a (nsp2-3 cleavage site region) were used. Samples were assayed by TaqMan multiplex real-time PCR using TaqMan Universal Master Mix II and a CFX384 Touch real-time PCR detection system (Bio-Rad). A standard curve was obtained using an in vitro transcript derived from a synthetic plasmid that contained all PCR targets. Each RNA sample was analyzed in triplicate.

## Histology and semi-quantitative scoring of lung pathology

Lungs were dissected from mice euthanized by intraperitoneal sodium pentobarbital injection as described above. The left lung lobe of each animal was removed at necropsy, instilled intratracheally with 4% paraformaldehyde (PFA), and post-fixed for 24 h by immersion in 4% PFA, prior to transfer into 70% ethanol and storage at 4 °C. Lung samples were then routinely processed to paraffin blocks and 5-μm thick serial sections prepared by cutting through the whole paraffin block and mounting sections 1:10 onto glass slides. These slides were stained with hematoxylin and eosin according to standard procedure and examined microscopically by an experienced veterinary pathologist. Microscopic lung findings were scored semi-quantitatively following accepted principles[74]. Briefly, all serial sections per animal were first evaluated blinded to treatment at low magnification (×40 to ×100) to select the lung level(s) with the most severe and extensive lesions for each animal. The extent of the lesions across this section was then estimated (and scored as 0 = < 5%; 1 = 5–33%; 2 = 33–66%; 3 = >66%) and this parameter taken as a weighing factor to multiply with the sum of all the individually scored lesions, calculating an overall combined lung pathology score per animal. The individual lesions were scored at high magnification (×200–400) and these included alveolar interstitial inflammatory cells (neutrophils, macrophages, and/or lymphocytes/plasma cells), perivascular mixed inflammatory cell infiltrate and edema, necrosis, intra-alveolar neutrophils, macrophages, and/or hemorrhage (each scored as 0 = none; 1 = mild, 2 = moderate, 3 = severe). Alveolar septal thickening (scored as 0 = none, 1 = 2-fold increase, 2 = 2–4-fold increase, 3 = more than 4-fold increase compared with unaffected septa), hyaline membranes, and intra-alveolar proteinaceous fluid (the latter two scored as 0 = none, 1 = 1, 2 = more than 1 per alveolus) were scored as well.

## Wild-type MERS-CoV neutralization assay

Huh7 (JCRB, No. JCRB0403) cells were seeded at a density of 12,000 cells/well in 96-well tissue culture plates 1 day prior to infection. Heat-inactivated (30 min at 56 °C) serum samples were analyzed in duplicate. The panel of sera were twofold serially diluted in duplicate, with an initial dilution of 1:10 and a final dilution of 1:1280 in 60 μL EMEM medium supplemented with penicillin, streptomycin, 2 mM L-glutamine and 2% FCS. Diluted sera were mixed with equal volumes of 120 $TCID_{50}$/60 μL rMERS-CoV and incubated for 1 h at 37 °C. The virus-serum mixtures were then transferred onto Huh7 cell monolayers and incubated at 37 °C and 5% $CO_2$. Cells either unexposed to the virus or mixed with 120 $TCID_{50}$/60 μL MERS-CoV were used as negative (uninfected) and positive (infected) controls, respectively. At 3 days post infection, cells were fixed and inactivated with 40 μL 37% formaldehyde/PBS solution/well overnight at 4 °C. Fixative was removed from the cells and the clusters were stained with 50 μL crystal violet solution per well, incubated for 10 min, and then rinsed off with water. Dried plates were evaluated for viral cytopathic effect. The virus neutralization titer was expressed as the reciprocal value of the highest dilution of the serum, which still inhibited virus replication. A rMERS-CoV backtitration was included with each assay run to confirm that the dose of the used inoculum was within the acceptable range of 30–300 $TCID_{50}$.

## Enzyme-linked immunosorbent assay (ELISA)

Clarified lung homogenates used for virus titration as described above were also used for determining levels of interferons and cytokines. The tested samples were diluted at 1:2 in diluent buffer provided for each ELISA kit and added to each well, and three multiple wells were set for each sample. ELISA kits specific for mouse IFN-β (DY8234-05, R&D Systems), TNF-α (DY410-05, R&D Systems), IL-6 (DY406-05, R&D Systems), IFN-λ(D485-05, R&D Systems), IFN-γ (DY1789B-05, R&D Systems), and IL-1β (D401-05, R&D Systems) were used according to the manufacturer's instructions.

## Passive serum transfer

Pooled serum for passive transfer experiments was obtained from hDPP4 KI mice (C57BL/6NTac-Dpp4tm3600(DPP4)Arte) that had been immunized 4 weeks prior with a single shot of DUB-negative rMERS-$CoV_{MA}$ ($10^4$ PFU) or mock-immunized with DMEM only. The murine antiserum was pooled from 15 mice and diluted 1:10 in PBS. Naive hDPP4 KI mice were passively immunized by intraperitoneal (i.p.) injection with 250 μL of DUB-negative rMERS-$CoV_{MA}$ or control DMEM serum. The following day, mice were challenged intranasally (i.n.) with a lethal dose of rMERS-$CoV_{MA}$ ($10^4$ PFU/mouse). Mice were sacrificed at day 2, 4, and 10 for virus lung titer determination as described above. Body weight and survival were monitored daily for 10 days and all mice showing more than 20% body weight loss, respiratory distress, or moribundity (humane endpoints) were euthanized as described above.

## ELISpot assay

IFN-γ ELISpot was performed on mouse splenocytes isolated from rMERS-$CoV_{MA-DUBneg}$- or mock-vaccinated hDPP4 KI mice (C57BL/6NTac-Dpp4tm3600(DPP4)Arte) at 4 weeks post immunization using a mouse IFN-γ ELISpot-plus kit (Mabtech). Splenocytes were obtained by mechanically dissociating spleens through a sterile cell strainer and restimulated for 18–20 h at 37 °C with a pool of 336 (168+168) peptides derived from a peptide scan (15-mers with 11-aa overlaps) through the MERS-CoV Spike glycoprotein (PM-MERS-CoV-S-1, JPT) at a final concentration of 1 μg/peptide/mL. Each sample of splenocytes was plated in triplicate wells and spots were developed using mouse IFN-γ ELISpot-plus kit (Mabtech), following the manufacturer's instructions. The total numbers of spots in each well were counted using an ELISpot reader and converted into the number of spots per 1 million splenocytes for each well. The medium/unstimulated splenocytes were used as negative controls and a CD3/CD28 mix (dilution 1:150) was used as a positive control. IFN-γ-secreting splenocytes were reported as the average of spot forming cells (SFCs) per million splenocytes for each sample.

## Statistical analysis

Statistical analysis was performed in the GraphPad Prism software package (GraphPad Software). Data are represented as means ± SEM of at least three replicative experiments unless otherwise stated. Statistical significance was analyzed with either Student's $t$ test, one-way analysis of variance (ANOVA), and the Tukey's post test. A two-way ANOVA was used where appropriate. P values of ****$P < 0.0001$, ***$P < 0.001$, **$P < 0.01$, and *$P < 0.05$ were considered significant. The number of animals in each experimental group and specific details on the statistical tests are included in the figure legends. NGS data analysis: quality control and trimmings of reads, Trimmomatic v0.36; mapping reads to the reference sequence (GenBank NC_019843.3), Samtools v1.7; variant calling: Bcftools v.1.7. Statistical analysis: GraphPad Prism 9.3.1.

## Reporting summary

Further information on research design is available in the Nature Portfolio Reporting Summary linked to this article.

## Data availability

The data that support the findings of this study are available in the article and supplementary information. Source data are provided with this paper. The raw NGS data generated in this study are deposited in SRA (BioProject: PRJNA933107, accession numbers: SRR23379932, rMERS-CoV v1691R Huh7 p10; SRR23379933, rMERS-CoV WT Huh7 p10) and can be accessed at this link: https://eur03.safelinks. protection.outlook.com/?url=https%3A%2F%2Fdataview.ncbi.nlm.nih. gov%2Fobject%2FPRJNA933107%3Freviewer%3Dfpbmt3cnd6620 pg1pjgjgtn1nr&data=05%7C01%7CI.Sidorov%40lumc.nl%7Cf297df6d 0c34449bf9b308db0b545aa4%7Cc4048c4fdd544cbd80495457aac d2fb8%7C0%7C0%7C638116229519224723%7CUnknown%7CTWF pbGZsb3d8eyJWIjoiMC4wLjAwMDAiLCJQIjoiV2luMzIiLCJBTiI6Ik1haW wiLCJXVCI6Mn0%3D%7C2000%7C%7C%7C&sdata=bhD06mLLy PjCJkjBqeF7TGuVdeXEgpYbTcQQ7pxndXg%3D&reserved=0.

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

## Acknowledgements

We are very grateful to the LUMC Experimental Animal facility for their support and we especially would like to thank Ewoud Speksnijder, Marleen Blom, Marloe Pijnacker-Verspuij and Jos van der Kaa. We thank Dr. Clara C. Posthuma and Jessika C. Zevenhoven-Dobbe (Leiden University Medical Center) for the initial modifications to pBAC-MERS-CoV. We also thank Dr. Ralf Bartenschlager (Heidelberg University) and Dr. Berend Jan Bosch (Utrecht University) for providing reagents, Ramon Arens, Iris N. Pardieck, and Yvonne de Vaal for sharing their expertise in analyzing T-cell responses and Dr. Ben A. Bailey-Elkin and Dr. Brian L. Mark (both University of Manitoba) for inspiring discussions. This research was performed as part of the Zoonoses Anticipation and Preparedness Initiative (ZAPI project; IMI Grant Agreement no. 115760) with

the assistance and financial support of IMI and the European Commission and in-kind contributions from EFPIA partners. This project has received funding from the European Union's Horizon 2020 research and innovation program under grant agreement No. 952373.

## Author contributions

S.K.M., P.J.B., R.C.M.K., E.J.S., and M.K. conceived research ideas and designed the experiments; S.K.M., P.J.B., R.C.M.K., T.J.D., S.T.M., M.E.L., S.Z.G., and S.A.L.Z. performed the experiments; N.O., L.E., and I.S. provided essential reagents and contributed to the DUB-negative MERS-CoV in vivo pilot studies; S.K.M., P.J.B., R.C.M.K., S.A.L.Z., I.A.S., E.J.S., and M.K. analyzed the data; S.K.M., E.J.S., and M.K. supervised the research; S.K.M., E.J.S., and M.K. wrote the paper. All authors reviewed and approved the manuscript.

## Competing interests

The authors declare no competing interests.
