## [Peer review file · Nature Communications]

REVIEWER COMMENTS

Reviewer #1 (Remarks to the Author):

This manuscript describes the development of a live-attenuated MERS vaccine candidate by one amino acid substitution (V1691R) of the viral protein PLpro rendering deubiquitinating enzyme (DUB) negative strain. While the mutation does not hinder viral growth in vitro, the mutant grows 2 log less in vivo and causes attenuated disease in hDPP4 mice. The virus is cleared in 14 days post infection and all animals survive with less lung pathology compared to the infection with wt rMERS-CoVMA. The authors also show that DUB-negative strain induces early innate immune activation in mice. Single intranasal immunization with the mutant elicits sterilizing neutralizing antibodies that prevent viral infection from lethal challenge. The authors indicate that the protection is largely due to neutralizing antibodies as passively transferred animals were also protected, though not fully.

The paper is well written, straightforward and the outcome data support the hypothesis and the conclusion. Here are my suggestions;

1. To my understanding, there are 2 set of rMERS-CoV one is based on EMC/2012 and the other one is based on a mouse-adapted strain, indicated as rMERS-CoVMA. The rMERS-CoV were tested on human cell lines and the mouse-adapted strain were tested on the mice. However, it is not clearly indicated that the DUB negative tested on mice model was rMERS-CoVMA-DUB. It was only indicated in the figures but missing indication of MA in the text. Therefore, it is a bit confusing and can be interpreted that the immunized DUB-negative has genetic background of the wt cell-adapted and not from mouse-adapted strain. An example;

Line 111: rMERS-CoV and the rMERS-CoVdubneg (DUB-negative MERS-CoV) then in line 115-114 rMERS-CoVMA and DUB-negative MERS-CoV is the same as non mouse-adapted strain. It will be clearer to readers if DUB-negative MERS-CoV with genetic background from mouse-adapted strain are indicated as DUB-negative MERS-CoVMA where applicable.

2. Only five passages are not enough to demonstrate genetic stability of a virus intended for use as a vaccine. Most LAV studies have performed at least P10, which is usually required in vaccine manufacturing process. Also, it was not clear whether only the region of the substitution (V1691R) was sequenced or the whole genome. The NGS data should be included in the supplementary figure to clearly demonstrate that there are no mutations in P5 and P10 both in substitution region and elsewhere in the genome that could affect the phenotype of the virus.

3. In this study, the role of ORF5 in reducing NF-KB was not mentioned, as the genetic background of all of rMERS-CoVs used here were from the cell-adapted EMC/2012 isolate and contain premature stop codon 108 in ORF5. The effect of DUB-negative virus in animal model observed here might not translate to human, especially knowing that complete deletion of ORF5 in MA strain increases virulence in mice (reference 36. Gutierrez-Alvarez, J., et al 2021)

4. An evidence of T-cell response stimulated by the DUB-negative mutant is missing in the manuscript. The passive immune transfer failed to afford complete protection, suggesting that T-cell responses also play an important role.

5. Line 134: the sentence “DUB-negative MERS-CoV was cleared at day 6” is not correlated with the graph as there were still 103 viral titers.

6. Line 248: 11 weeks is not quite qualified as long-lasting. I suggest to use “sustained” neutralizing antibodies to 11 weeks.

7. Fig 6D: please indicate the limit of detection on the graph.

8. Line 275, 279, and 281: There is no lung pathology pictures included in supplementary Fig. 4. It is important to see overall lung pathology of immunized animals rather than just the semi-quantitative lung pathology score as it can be subjective, particularly when DUB-negative strain itself gave some lung pathology to the immunized animal (figure 3).

9. Line 299: As previously mentioned, T cell responses data should give a clearer picture to the absence of full protection after passive transfer. It should be interesting to know how T-cell responses of a live attenuated vaccine compares to viral vectored MERS vaccines, MVA and ChAdOx1 MERS vaccines.

10. Line 379: It is highly likely (Chose one but as Nature Comm prefers us not to use highly or very. Maybe go with Likely or another synonymous word.)

11. Line 429-434: There is not enough data to state that “the mutation introduced in MERS-CoV PLpro has high stability against possible reversion” as the virus can mutate elsewhere in its genome and recover its virulence. Line 431; no NGS data of at least passages 10 to back up the statement.

12. The deletion of accessory proteins is to be cautioned as the previous MA strain that was engineered with full deletion of ORF5 was more virulent in mice than the early stop codon mutant. TRS mutation maybe safer in terms of reducing recombination events and improving the stability of the engineered virus.

13. Line 470-478: The protocol for the construction of rMERS-CoV is not quite clear about the genetic background of the virus used. Please consider rephrasing. Also, where applicable, briefly describe short protocols rather than just quoting references.

14. Line 741-743 and line 830-832: The reference numbers 34 and 35 are the same as reference numbers 71 and 72.

15. Line 850: growth curve figure 1A and 1B were performed only once.

16. Figure 3: Consistency of the order of the data demonstrated; rMERS-CoVMA-DUBneg (in red) was shown after rMERS-CoVMA (in blue) in previous figure 1 and 2. Changing the order to be the same as the previous figure will also correlate with fig 3A as well.

17. Figure 5B: The immunization is a single intranasal administration of the DUB-negative strain. Please explain why the neutralizing antibodies were increased at week 7 while the animals were not boosted.

18. Figure 6D shows complete sterilization of the inoculum virus as no infectious particles are detected in the lungs. However, the authors should perform qRT-PCR to determine the level of genomic viral RNA, which could give insight on viral inhibition in immunized animals.

Reviewer #2 (Remarks to the Author):

Myeni et al describe in vitro and in vivo properties of an attenuated construct of MERS-Cov they propose as a new candidate for a human vaccine. Previous work demonstrated that substitution in the ubiquitin binding site of MERS-Cov papain-like protease (PLpro) disrupt its deubiquitinating enzyme (DUB) activity. Authors confirm that this modification is not affecting viral replication dynamics in Huh7 and MRC5 human cell lines. However, the V1691R mutation abrogating DUB activity also result in reduced IFN- β promoter inhibition therefore restoring host cells innate response capacity. The attenuated phenotype of the DUB-negative MERS-Cov is confirmed in the DPP4 knock-in mouse model highly susceptible (100% mortality in about 5 days post-infection) to mouse adapted MERS-Cov. The V1691R attenuated virus have reduced replication capacity in the lung of challenged mice and accelerated clearance. In addition, mice infected with DUB-neg attenuated virus have significantly reduced lung pathology and none of the animals died from infection. Interesting, at day 1 pi, attenuated phenotype in mice is associated with increased Type I and Type III interferon response as well as increased IL-6 and TNF- α , in agreement the impaired capacity of the attenuated virus to counteract innate host response. Finally, a single intranasal immunization of mice with the DUB-negative attenuated virus (10⁴ pfu) induce strong neutralizing antibody response persisting up to 9-11 weeks. These nAb have broad activity against a diversity of MERS-Cov strains. Sterilizing immunity was obtained against lethal MERS-Cov challenge and passive transfer study in mice demonstrated that neutralizing antibodies contribute to a significant part of observed protection.

To conclude, this is a very well described study with strong methodology convincing on the attenuated phenotype of the DUB-negative virus in cell cultures and in a the mice model. However, in the perspective of developing a modified attenuate vaccine against MERS-Cov there are different points which deserves further characterization or at least needed to be more detailed in the discussion sections:

1. The authors demonstrated that no reversion of V1691R mutation was observed over five passages in cell culture and 14 days post infection in mice. This is an important information but certainly limited. Prolonged follow-up, with serial passages in vivo and co-infection studies would be required in order to guarantee the long-term genetic stability of the vaccines or absence of risk of complementation by wild type viruses.
2. Circulation among individuals of the attenuated vaccine may be of concern, especially in populations with increased vulnerabilities, like immuno-compromised patients. This may be assessed in transmission studies in animal models.
3. Attenuation in mice was assessed at intermediate challenge doses. What would be the outcome in animals exposed to high doses of the DUB-neg vaccine (i.e; 10⁶ or 10⁸ pfu)?

Roger Le Grand

Reviewer #1 (Remarks to the Author):

This manuscript describes the development of a live-attenuated MERS vaccine candidate by one amino acid substitution (V1691R) of the viral protein PLpro rendering deubiquitinating enzyme (DUB) negative strain. While the mutation does not hinder viral growth in vitro, the mutant grows 2 log less in vivo and causes attenuated disease in hDPP4 mice. The virus is cleared in 14 days post infection and all animals survive with less lung pathology compared to the infection with wt rMERS-CoVMA. The authors also show that DUB-negative strain induces early innate immune activation in mice. Single intranasal immunization with the mutant elicits sterilizing neutralizing antibodies that prevent viral infection from lethal challenge. The authors indicate that the protection is largely due to neutralizing antibodies as passively transferred animals were also protected, though not fully.

The paper is well written, straightforward and the outcome data support the hypothesis and the conclusion. Here are my suggestions;

We thank the reviewer for his/her very positive evaluation and valuable suggestions.

1. To my understanding, there are 2 set of rMERS-CoV one is based on EMC/2012 and the other one is based on a mouse-adapted strain, indicated as rMERS-CoVMA. The rMERS-CoV were tested on human cell lines and the mouse-adapted strain were tested on the mice. However, it is not clearly indicated that the DUB negative tested on mice model was rMERS-CoVMA-DUB. It was only indicated in the figures but missing indication of MA in the text. Therefore, it is a bit confusing and can be interpreted that the immunized DUB-negative has genetic background of the wt cell-adapted and not from mouse-adapted strain. An example:

Line 111: rMERS-CoV and the rMERS-CoVdubneg (DUB-negative MERS-CoV) then in line 115-114 rMERS-CoVMA and DUB-negative MERS-CoV is the same as non-mouse-adapted strain. It will be

clearer to readers if DUB-negative MERS-CoV with genetic background from mouse-adapted strain are indicated as DUB-negative MERS-CoV_{MA} where applicable.

The reviewer is correct, indeed there are two sets of rMERS-CoVs used in this study, one based on the EMC/2012 isolate and the other one based on a mouse-adapted MERS-CoV. All the recombinant viruses used in this study including the rMERS-CoV and rMERS-CoV_{DUBneg}, and the mouse-adapted viruses (rMERS-CoV_{MA} and rMERS-CoV_{MA-DUBneg}) were tested in human cell lines, see **Fig. 1A, B and Supplementary Fig. 1A, B**. The mouse-adapted viruses were further tested in human DPP4 KI mice. We thank the reviewer for pointing out that we should have made the textual distinction between these viruses more consistent/clear. To improve clarity, we have now changed the “DUB-negative MERS-CoV” with the genetic background from a mouse-adapted strain to “DUB-negative rMERS-CoV_{MA}”, where appropriate in the text.

2. Only five passages are not enough to demonstrate genetic stability of a virus intended for use as a vaccine. Most LAV studies have performed at least P10, which is usually required in vaccine manufacturing process. Also, it was not clear whether only the region of the substitution (V1691R) was sequenced or the whole genome. The NGS data should be included in the supplementary figure to clearly demonstrate that there are no mutations in P5 and P10 both in substitution region and elsewhere in the genome that could affect the phenotype of the virus.

We agree with reviewer 1 that five rounds of passaging in cell culture is limited for a virus intended for use as a vaccine. As discussed in the manuscript, our studies provide a proof-of-concept for the design of MLV coronavirus vaccines based on the selective inactivation of their PLpro DUB activity. Further development of the attenuated DUB-negative MERS-CoV would indeed have to include an extensive analysis of the genetic stability of the mutant virus in both cell culture (for manufacturing purpose) and *in vivo* (for safety assessment). The incorporation of some of the strategies discussed

in the manuscript might further improve the safety profile and stability of a candidate live vaccine. In particular, a combination of DUB-inactivating mutations could be developed to minimize the problem of (pseudo)reversion, but we consider such studies clearly beyond the scope of the present manuscript.

However, in order to explore reviewer 1's concern, and develop a first impression of the genetic stability of the introduced mutation in cell culture, we have now passaged both the wt (rMERS-CoV) and the DUB-negative rMERS-CoV in Huh7 cells 10 times. A summary of the NGS data is now included in **Supplementary Table 1**. We have adjusted the text in the Results/discussion section to accommodate this work and the conclusions (**Lines 116-130**). The NGS analysis of the full genome of the DUB-negative rMERS-CoV P10 virus indicate that 62% of the sequences still show the originally introduced DUB-inactivating mutation, and the variations seen at the mutated codon were predominantly substitutions to C, S or H, each requiring a single nt substitution (GTG = Val; CAC = His; AGC = Ser and TGC = Cys) and each occurring with a frequency of around 10%. Importantly, full reversion to a V codon (requiring 2 nt substitutions) could not be detected. We also found some low frequency mutations (mostly below 10% of the population) in other regions of the viral genome sequence.

3. In this study, the role of ORF5 in reducing NF-KB was not mentioned, as the genetic background of all of rMERS-CoVs used here were from the cell-adapted EMC/2012 isolate and contain premature stop codon 108 in ORF5. The effect of DUB-negative virus in animal model observed here might not translate to human, especially knowing that complete deletion of ORF5 in MA strain increases virulence in mice (reference 36. Gutierrez-Alvarez, J., et al 2021)

A premature stop codon 108 in ORF5 was deliberately introduced in all rMERS-CoVs used in this study to avoid complications during virus passaging due to ORF5 evolution and associated changes in

host immune suppression in cell culture systems and mouse lungs (reference 34, 35, 36, Menachery, V.D, et al 2017 and also mentioned and discussed in **Materials and Methods, Lines 499-506**).

While the absence of ORF5 may enhance pathogenesis in mice (reference 36), the mouse-adapted parental virus (rMERS-CoV_{MA}) used to generate the DUB-negative virus causes a lethal lung disease in mice while the DUB-negative virus clearly is strongly attenuated.

ORF5 has been reported to be highly stable *in vivo* in both human and camel isolates (reference 36). Future studies in camels utilizing wt and DUB-negative viruses that express full-length ORF5 or which have a complete deletion of ORF5 or a premature stop codon 108 in ORF5 (used in this study), might shed light on whether the effect of the DUB-negative virus seen in this mouse model can translate to camels, where ORF5 - as in humans - seems to be stable. Since we think this is beyond the scope of this work, we have not further elaborated on it in the manuscript. We did add a note on ORF5 in the discussion to address this (reviewer's point 12), please see below.

4. An evidence of T-cell response stimulated by the DUB-negative mutant is missing in the manuscript. The passive immune transfer failed to afford complete protection, suggesting that T-cell responses also play an important role.

We thank the reviewer for this useful suggestion and have now performed an additional animal experiment to test the effect of DUB-negative rMERS-CoV_{MA} vaccination on T-cell immunity in hDPP4 KI mice, while comparing with mock-vaccinated mice. Using an IFN- γ ELISpot assay, we measured splenic T-cell responses against a pool of peptides spanning the complete spike protein sequence (see **Material and Methods, Lines 663-677**). At four weeks post-vaccination, mice immunized with the DUB-negative rMERS-CoV_{MA} elicited significantly higher levels MERS-CoV spike specific IFN- γ producing T cells compared to the mock vaccinated animals. This data shows that the DUB-negative virus elicited S-specific cellular responses in mice (**Supplementary Fig. 5**) and suggest that T-cell

responses together with neutralizing antibodies may indeed also play an important role in protection against MERS-CoV (**Lines 445-452**).

5. Line 134: the sentence “DUB-negative MERS-CoV was cleared at day 6” is not correlated with the graph as there were still 103 viral titers.

The original sentence did not state that the “ DUB-negative MERS-CoV was cleared at day 6”. Please see the original sentence below. However, to avoid any confusion, we have now modified the sentence as shown below.

Original sentence: Furthermore, DUB-negative rMERS-CoV_{MA} was cleared from the lungs and at day 6 p.i. lung virus titers for the modified virus had significantly decreased to $\sim 1 \times 10^3$ PFU per g of lung tissue for 50% of the animals, while no progeny was measured for the other 50% of the animals at that time point (**Fig. 1D**).

Changed sentence (Line 142): Furthermore, over time the DUB-negative rMERS-CoV_{MA} was cleared from the lungs. At day 6 p.i., lung virus titers for the modified virus had significantly decreased to $\sim 1 \times 10^3$ PFU per g of lung tissue for 50% of the animals, while no progeny was measured for the other 50% of the animals at that time point (**Fig. 1D**).

6. Line 248: 11 weeks is not quite qualified as long-lasting. I suggest to use “sustained” neutralizing antibodies to 11 weeks.

Where appropriate in the text, we have changed the term “long-lasting” to “sustained” as suggested by the reviewer.

7. Fig 6D: please indicate the limit of detection on the graph.

We have included the limit of detection for infectious viral progeny titers, which is now indicated with a dashed line. See **Fig. 6D** and also where relevant for other figures.

8. Line 275, 279, and 281: There is no lung pathology pictures included in supplementary Fig. 4. It is important to see overall lung pathology of immunized animals rather than just the semi-quantitative lung pathology score as it can be subjective, particularly when DUB-negative strain itself gave some lung pathology to the immunized animal (figure 3).

We have used the semi-quantitative lung pathology scores aiming to capture the severity and extent of the observed lung lesions in an unbiased and reproducible manner across all animals from both experiments (as explained in the methods section). Nevertheless, we agree with reviewer 1 that it may help to add some photomicrographs of lungs with representative lesions in **Supplementary Fig. 4**, to better appreciate the differences between mock and DUB-negative rMERS-CoV_{MA} immunized animals.

We have now added two such photomicrographs in **Supplementary Fig. 4A** and moved the bar chart to **Supplementary Fig. 4B**. To accommodate this change further in the Results section, we have also adjusted the text in **Lines 286-297**

9. Line 299: As previously mentioned, T cell responses data should give a clearer picture to the absence of full protection after passive transfer. It should be interesting to know how T-cell responses of a live attenuated vaccine compares to viral vectored MERS vaccines, MVA and ChAdOx1 MERS vaccines.

As mentioned in the response to the previous comment (No. 4), we performed an additional animal experiment to evaluate whether the DUB-negative rMERS-CoV_{MA} is capable of inducing cellular responses by 4 weeks post vaccination. Our data show that a single-dose of the DUB-negative rMERS-CoV_{MA} induces cellular responses in mice (**Supplementary Fig. 5**). We agree with reviewer 1 that it will be very interesting to learn how the T-cell responses induced by a live attenuated vaccine, like the DUB-negative rMERS-CoV_{MA}, compare to those induced by virally vectored MERS vaccines, like the MVA and ChAdOx1 MERS vaccines. However, in our opinion, such an elaborate comparison

is clearly beyond the scope of this study, and will need to be the focus of future experiments aiming to compare the (humoral and cellular) immunogenicity of these vaccines side by side in the same animal model. As mentioned in the Discussion section of the original manuscript, live attenuated vaccines often induce excellent immune responses (humoral and cellular) and often provide lifelong immunity. Similar to a natural infection with MERS-CoV, live attenuated MERS-CoV vaccines are expected to induce broad T cell and humoral immune responses. The ability to intranasally deliver live attenuated vaccines like the DUB-negative MERS-CoV may provide a major advantage due to enhanced mucosal immunity compared to other vaccines, including MERS-CoV vectored candidate vaccines, which are mostly administered intramuscularly. Live attenuated vaccines also induce immunity against a range of MERS-CoV proteins, in addition to the spike proteins used in vectored vaccines, thus providing additional viral epitopes.

10. Line 379: It is highly likely (Chose one but as Nature Comm prefers us not to use highly or very. Maybe go with Likely or another synonymous word.)

We have taken reviewer 1's feedback and have changed "highly likely" to just "likely" in now **Line 395** in the text and where fitting elsewhere in the manuscript.

11. Line 429-434: There is not enough data to state that "the mutation introduced in MERS-CoV PLpro has high stability against possible reversion" as the virus can mutate elsewhere in its genome and recover its virulence. Line 431; no NGS data of at least passages 10 to back up the statement.

Line 450: We agree with the reviewer that there is not enough data to state that " the mutation introduced in MERS-CoV PLpro has high stability against possible reversion". We have explored reviewer 1's concern about the genetic stability of the introduced mutation in cell culture, where both the wt rMERS-CoV and the DUB-negative rMERS-CoV viruses were passaged 10 times in Huh7 cells and analyzed by NGS (see also our response to point **No. 2**). It is important to stress that after 10 passages, and in spite of some evolution occurring at the mutated codon 838/1691, full reversion

had not occurred and no variants have emerged that have become dominant in the population, with most abundant (non-synonymous) single-nt variations observed not exceeding 20.3% (Data included in **Supplementary Table S1**).

We have now modified the text in our Discussion (**Lines 450-452**) and changed our statement to reflect the initial results from the full-genome sequencing of the P10 viruses (wt rMERS-CoV and the DUB-negative rMERS-CoV p10).

12. The deletion of accessory proteins is to be cautioned as the previous MA strain that was engineered with full deletion of ORF5 was more virulent in mice than the early stop codon mutant. TRS mutation maybe safer in terms of reducing recombination events and improving the stability of the engineered virus.

We agree with reviewer 1 that the deletion of accessory proteins is to be cautioned as the previous mouse adapted virus engineered with full deletion of ORF5 was more virulent in mice (reference 36). We have now made that clear in our discussion and have discussed the re-engineering of viral transcription regulatory sequences as a strategy to reduce the likelihood of successful recombination with other coronaviruses other than MERS-CoV and improving the stability of the engineered virus (now **Lines 460-465**).

13. Line 470-478: The protocol for the construction of rMERS-CoV is not quite clear about the genetic background of the virus used. Please consider rephrasing. Also, where applicable, briefly describe short protocols rather than just quoting references.

Recombinant rMERS-CoV and rMERS-CoV_{DUBneg} were derived from a MERS-CoV full-length cDNA clone based on MERS-CoV strain EMC/2012 (reference 73). The recombinant mouse-adapted viruses rMERS-CoV_{MA} and rMERS-CoV_{MA-DUBneg} were generated from the pBAC-MERS^{FL}-MA-5FL full-length cDNA clone (reference 36) that was based on the mouse-adapted MERS_{MA}6.1.2 virus (reference 35).

As Nature Communications requires that the Methods section be written as concisely as possible, we have now added the information about the genetic background of the viruses used in **Materials and Methods (Line 494-497)** by including the references in which they have been extensively described.

14. Line 741-743 and line 830-832: The reference numbers 34 and 35 are the same as reference numbers 71 and 72.

We apologize for this mistake and have now deleted reference numbers 71 and 72, and adjusted the referencing in the text.

15. Line 850: growth curve figure 1A and 1B were performed only once.

The growth curve in **Fig. 1A and 1B** was performed twice (now also made clear in the figure legend), however the replication of all viruses used in this study were evaluated at specified time points at least three times and statistical comparisons performed (see also **Supplementary Fig. 1**).

16. Figure 3: Consistency of the order of the data demonstrated; rMERS-CoV_{MA-DUBneg} (in red) was shown after rMERS-CoV_{MA} (in blue) in previous figure 1 and 2. Changing the order to be the same as the previous figure will also correlate with fig 3A as well.

We apologize for this inconsistency of the order of the data and where feasible in all the figures, we have now kept the order of the rMERS-CoV_{MA-DUBneg} (shown in red) and the wt rMERS-CoV_{MA} (shown in blue) consistent.

17. Figure 5B: The immunization is a single intranasal administration of the DUB-negative strain.

Please explain why the neutralizing antibodies were increased at week 7 while the animals were not boosted.

Yes, the immunization is a single intranasal administration of the DUB-negative strain, however as the reviewer has pointed out the neutralizing antibody levels appear to be increased at week 7 while

the animals were not boosted. The neutralizing antibody response seems to increase from week 2 to 4, then remain more or less constant between week 4 and 11 (**Supplementary Fig. 3**). The neutralization assays/runs were performed independently for each specified time point post vaccination and not in the same experiment/run. So the modest increase at week 7 (**Fig. 5B**) and the drop at week 9 (**Supplementary Fig. 3**) could be a result of experiment-to-experiment variation and do not necessarily reflect a change in neutralizing antibody levels in the vaccinated animals. We did not have a known standard neutralizing antibody with a known/established titer to control our experiments unfortunately. We did however include the virus back titration in each run to confirm the dose of virus inoculum. The assay used is very sensitive (IC_{100}), CPE-based and less effective in providing more quantitative measurements of the antibody strength. The acceptable back titration range of 30 to 300 $TCID_{50}$ is also quite wide. While we targeted to use a dose of 120 $TCID_{50}/60 \mu L$ MERS-CoV, the back titration titer of the dose used in week 7 was 55 $TCID_{50}/60 \mu L$ compared to 190 $TCID_{50}/60 \mu L$ MERS-CoV in week 9, which might explain measuring increased neutralizing antibody levels at week 7 and decreased neutralizing titers at week 9. Due to the limited amounts of sera recovered from immunized animals for each time point, it was not possible to repeat the assay for all the samples (from different time points) in one single run for direct comparison. Moreover, it was necessary to perform the experiments for specified time points in independent runs as some time points before challenge with the lethal rMERS-CoV_{MA} (week 4 to 7) served as Go/No Go decision determinants requested by our animal welfare department. We have now added a note in our discussion to address the reviewer's concern (**Lines 422-425**).

18. Figure 6D shows complete sterilization of the inoculum virus as no infectious particles are detected in the lungs. However, the authors should perform qRT-PCR to determine the level of genomic viral RNA, which could give insight on viral inhibition in immunized animals.

We have now performed the requested qRT-PCR analysis to determine the levels of genome (ORF1a) and sub-genomic (N gene) viral RNA in the lungs of DUB-negative rMERS-CoV_{MA} vaccinated mice at

day 0, 2, 4, 6 and 14 after challenge with a lethal dose of rMERS-CoV_{MA} (Materials and Methods, **Lines 592-602**). On day 2 and 4, the lungs of mock-vaccinated animals had high MERS-CoV RNA levels (**Supplementary Fig. 6, Lines 280-285**) consistent with the high levels of infectious virus (**Fig. 6D**). In contrast, DUB-negative rMERS-CoV_{MA}-vaccinated mice had significantly reduced viral RNA levels (at least 5 log reduction) in the lungs at day 2 and 4 post challenge, which were similar to viral RNA levels at day 0, 6 and 14 (**Supplementary Fig. 6, Lines 280-285**). This data is consistent with the undetectable levels of infectious virus in the lungs of vaccinated animals (**Fig. 6D**) demonstrating the complete sterilization of the lethal dose of wt virus in DUB-negative rMERS-CoV_{MA} immunized animals.

Reviewer #2 (Remarks to the Author):

Myeni et al describe in vitro and in vivo properties of an attenuated construct of MERS-Cov they propose as a new candidate for a human vaccine. Previous work demonstrated that substitution in the ubiquitin binding site of MERS-Cov papain-like protease (PLpro) disrupt its deubiquitinating enzyme (DUB) activity. Authors confirm that this modification is not affecting viral replication dynamics in Huh7 and MRC5 human cell lines. However, the V1691R mutation abrogating DUB activity also result in reduced IFN- β promoter inhibition therefore restoring host cells innate response capacity. The attenuated phenotype of the DUB-negative MERS-Cov is confirmed in the DPP4 knock-in mouse model highly susceptible (100% mortality in about 5 days post-infection) to mouse adapted MERS-Cov. The V1691R attenuated virus have reduced replication capacity in the lung of challenged mice and accelerated clearance. In addition, mice infected with DUB-neg attenuated virus have significantly reduced lung pathology and none of the animals died from infection. Interesting, at day 1 pi, attenuated phenotype in mice is associated with increased Type I and Type III interferon response as well as increased IL-6 and TNF- α , in agreement the impaired capacity of the attenuated virus to counteract innate host response. Finally, a single intranasal

immunization of mice with the DUB-negative attenuated virus (10⁴ pfu) induce strong neutralizing antibody response persisting up to 9-11 weeks. These nAb have broad activity against a diversity of MERS-Cov strains. Sterilizing immunity was obtained against lethal MERS-Cov challenge and passive transfer study in mice demonstrated that neutralizing antibodies contribute to a significant part of observed protection.

To conclude, this is a very well described study with strong methodology convincing on the attenuated phenotype of the DUB-negative virus in cell cultures and in a the mice model. However, in the perspective of developing a modified attenuate vaccine against MERS-Cov there are different points which deserves further characterization or at least needed to be more detailed in the discussion sections:

We thank the reviewer for his very positive evaluation and we appreciate the valuable suggestions.

1. The authors demonstrated that no reversion of V1691R mutation was observed over five passages in ell culture and 4 days post infection in mice. This is an important information but certainly limited. Prolonged follow-up, with serial passages *in vivo* and co-infection studies would be required in order to guarantee the long-term genetic stability of the vaccines or absence of risk of complementation by wild type viruses.

Please refer to **Reviewer #1, point 2 and 11**. We also agree with reviewer 2 that ideal future experiments (beyond the scope of this study) will have to look into prolonged follow-up serial passages *in vivo*, NGS analysis of p10 or higher viruses from *in vivo* passaging with statistical replicates, and the incorporation of strategies discussed in the manuscript to improve the safety profile and stability of the live attenuated DUB-negative virus to guarantee the long-term genetic stability of the DUB-negative MERS-CoV.

2. Circulation among individuals of the attenuated vaccine may be of concern, especially in

populations with increased vulnerabilities, like immuno-compromised patients. This may be assessed in transmission studies in animal models.

We agree with reviewer 2 that the circulation of modified live attenuated viruses like the DUB-negative MERS-CoV may be of concern in populations with increased vulnerabilities. Live attenuated vaccine candidates like the DUB-negative virus administered intranasally might however, have an advantage of priming early protective innate cellular responses as well as inducing localized adaptive memory through subsequent infection of the upper respiratory tract (also mentioned in **Lines 431-437**). As a result, this localized adaptive memory would significantly reduce/abrogate disease and symptoms therefore prevent or considerably reduce transmission to high-risk populations or immunocompromised patients. This of course would have to be assessed in the future in appropriate animal transmission models for MERS-CoV. Alternatively, the DUB-negative rMERS-CoV could be engineered to allow for single cycle replication without spread in a trans-complementation system (replicon RNAs in cells that express missing genes like structural genes in the replicon), recently demonstrated for SARS-CoV-2 by Inna Ricardo-Lax, et al Science, 2021.

3. Attenuation in mice was assessed at intermediate challenge doses. What would be the outcome in animals exposed to high doses of the DUB-neg vaccine (i.e; 10^6 or 10^8 pfu)?

While it would be interesting to explore the influence of high doses of the attenuated DUB-negative virus in mice, this was not possible because both the wt rMERS-CoV_{MA} and the DUB-negative rMERS-CoV_{MA} viruses grew to similar but lower titers (between 5.2×10^6 to 8.8×10^6 PFU/mL) in human and primary mouse cell cultures. Thus it was not feasible to get to 10^6 or 10^8 PFU/50uL intranasal dose (maximum volume allowed for intranasal administration in mice is 50uL). It is important to emphasize that the doses (10^4 to 10^5) used in this study of the wt rMERS-CoV_{MA} virus are highly lethal in mice (see **Supplementary Fig. 2**). An inoculum of 5×10^3 PFU was established as the lethal dose in hDPP4 KI mice (reference 35). It is also important to emphasize that even though high infectious virus loads of the DUB-negative rMERS-CoV_{MA} virus ($>10^6$ PFU/g lung) were recovered after 1-2 days

of infection in mice, all animals infected with the DUB-negative virus survived and, like the mock-infected animals, showed no signs of morbidity and kept a relatively stable body weight (see also **Fig. 1 and Fig. 2**).

REVIEWERS' COMMENTS

Reviewer #1 (Remarks to the Author):

The Reviewer would like to thank the authors for considering the points made and the extra data provided which enhance the clarity of the content of this manuscript. Most points were clearly addressed, only a couple of small questions that may be discussed.

L127: V383C,S,H mutants would still render the same DUB-negative phenotype?

L297: A typo of (“) Before -Taken together.

Supplementary figure 6A and B were done from the same samples used in figure 6. The viral RNA detected $\sim 10^{11}$ or 10^{12} copies were correlated to $\sim 10^7$ - 10^8 infectious particles however when the viral RNA detected at 10^6 RNA copies, this does not confer at least 10^1 - 10^2 particles which would be within the limit of detection (10 PFU). Would you maybe discuss or speculate on this discrepancy?

Reviewer Comments (Remarks to the Author)

Reviewer #1 (Remarks to the Author):

This manuscript describes the development of a live-attenuated MERS vaccine candidate by one amino acid substitution (V1691R) of the viral protein PLpro rendering deubiquitinating enzyme (DUB) negative strain. While the mutation does not hinder viral growth in vitro, the mutant grows 2 log less in vivo and causes attenuated disease in hDPP4 mice. The virus is cleared in 14 days post infection and all animals survive with less lung pathology compared to the infection with wt rMERS-CoVMA. The authors also show that DUB-negative strain induces early innate immune activation in mice. Single intranasal immunization with the mutant elicits sterilizing neutralizing antibodies that prevent viral infection from lethal challenge. The authors indicate that the protection is largely due to neutralizing antibodies as passively transferred animals were also protected, though not fully.

The paper is well written, straightforward and the outcome data support the hypothesis and the conclusion. Here are my suggestions;

1. To my understanding, there are 2 set of rMERS-CoV one is based on EMC/2012 and the other one is based on a mouse-adapted strain, indicated as rMERS-CoVMA. The rMERS-CoV were tested on human cell lines and the mouse-adapted strain were tested on the mice. However, it is not clearly indicated that the DUB negative tested on mice model was rMERS-CoVMA-DUB. It was only indicated in the figures but missing indication of MA in the text. Therefore, it is a bit confusing and can be interpreted that the immunized DUB-negative has genetic background of the wt cell-adapted and not from mouse-adapted strain. An example:
Line 111: rMERS-CoV and the rMERS-CoVdubneg (DUB-negative MERS-CoV) then in line 115-114 rMERS-CoVMA and DUB-negative MERS-CoV is the same as non mouse-adapted strain. It will be clearer to readers if DUB-negative MERS-CoV with genetic background from mouse-adapted strain are indicated as DUB-negative MERS-CoVMA where applicable.
2. Only five passages are not enough to demonstrate genetic stability of a virus intended for use as a vaccine. Most LAV studies have performed at least P10, which is usually required in vaccine manufacturing process. Also, it was not clear whether only the region of the substitution (V1691R) was sequenced or the whole genome. The NGS data should be included in the supplementary figure to clearly demonstrate that there are no mutations in P5 and P10 both in substitution region and elsewhere in the genome that could affect the phenotype of the virus.
3. In this study, the role of ORF5 in reducing NF-KB was not mentioned, as the genetic background of all of rMERS-CoVs used here were from the cell-adapted EMC/2012 isolate and contain premature stop codon 108 in ORF5. The effect of DUB-negative virus in animal model observed here might not translate to human, especially knowing that complete deletion of ORF5 in MA strain increases virulence in mice (reference 36. Gutierrez-Alvarez, J., et al 2021)
4. An evidence of T-cell response stimulated by the DUB-negative mutant is missing in the manuscript. The passive immune transfer failed to afford complete protection, suggesting that T-cell responses also play an important role.
5. Line 134: the sentence "DUB-negative MERS-CoV was cleared at day 6" is not correlated with the graph as there were still 103 viral titers.
6. Line 248: 11 weeks is not quite qualified as long-lasting. I suggest to use "sustained" neutralizing

antibodies to 11 weeks.

7. Fig 6D: please indicate the limit of detection on the graph.

8. Line 275, 279, and 281: There is no lung pathology pictures included in supplementary Fig. 4. It is important to see overall lung pathology of immunized animals rather than just the semi-quantitative lung pathology score as it can be subjective, particularly when DUB-negative strain itself gave some lung pathology to the immunized animal (figure 3).

9. Line 299: As previously mentioned, T cell responses data should give a clearer picture to the absence of full protection after passive transfer. It should be interesting to know how T-cell responses of a live attenuated vaccine compares to viral vectored MERS vaccines, MVA and ChAdOx1 MERS vaccines.

10. Line 379: It is highly likely (Chose one but as Nature Comm prefers us not to use highly or very. Maybe go with Likely or another synonymous word.)

11. Line 429-434: There is not enough data to state that “the mutation introduced in MERS-CoV PLpro has high stability against possible reversion” as the virus can mutate elsewhere in its genome and recover its virulence. Line 431; no NGS data of at least passages 10 to back up the statement.

12. The deletion of accessory proteins is to be cautioned as the previous MA strain that was engineered with full deletion of ORF5 was more virulent in mice than the early stop codon mutant. TRS mutation maybe safer in terms of reducing recombination events and improving the stability of the engineered virus.

13. Line 470-478: The protocol for the construction of rMERS-CoV is not quite clear about the genetic background of the virus used. Please consider rephrasing. Also, where applicable, briefly describe short protocols rather than just quoting references.

14. Line 741-743 and line 830-832: The reference numbers 34 and 35 are the same as reference numbers 71 and 72.

15. Line 850: growth curve figure 1A and 1B were performed only once.

16. Figure 3: Consistency of the order of the data demonstrated; rMERS-CoVMA-DUBneg (in red) was shown after rMERS-CoVMA (in blue) in previous figure 1 and 2. Changing the order to be the same as the previous figure will also correlate with fig 3A as well.

17. Figure 5B: The immunization is a single intranasal administration of the DUB-negative strain. Please explain why the neutralizing antibodies were increased at week 7 while the animals were not boosted.

18. Figure 6D shows complete sterilization of the inoculum virus as no infectious particles are detected in the lungs. However, the authors should perform qRT-PCR to determine the level of genomic viral RNA, which could give insight on viral inhibition in immunized animals.

Reviewer #2 (Remarks to the Author):

Myeni et al describe in vitro and in vivo properties of an attenuated construct of MERS-Cov they propose as a new candidate for a human vaccine. Previous work demonstrated that substitution in the ubiquitin binding site of MERS-Cov papain-like protease (PLpro) disrupt its deubiquitinating enzyme (DUB) activity. Authors confirm that this modification is not affecting viral replication dynamics in Huh7 and MRC5 human cell lines. However, the V1691R mutation abrogating DUB activity also result in reduced IFN- β promoter inhibition therefore restoring host cells innate response capacity. The attenuated phenotype of the DUB-negative MERS-Cov is confirmed in the DPP4 knock-in mouse model highly susceptible (100% mortality in about 5 days post-infection) to mouse adapted MERS-Cov. The V1691R attenuated virus have reduced replication capacity in the lung of challenged mice and accelerated clearance. In addition, mice infected with DUB-neg attenuated virus have significantly reduced lung pathology and none of the animals died from infection. Interesting, at day 1 pi, attenuated phenotype in mice is associated with increased Type I and Type III interferon response as well as increased IL-6 and TNF- α , in agreement the impaired capacity of the attenuated virus to counteract innate host response. Finally, a single intranasal immunization of mice with the DUB-negative attenuated virus (10⁴ pfu) induce strong neutralizing antibody response persisting up to 9-11 weeks. These nAb have broad activity against a diversity of MERS-Cov strains. Sterilizing immunity was obtained against lethal MERS-Cov challenge and passive transfer study in mice demonstrated that neutralizing antibodies contribute to a significant part of observed protection.

To conclude, this is a very well described study with strong methodology convincing on the attenuated phenotype of the DUB-negative virus in cell cultures and in a the mice model. However, in the perspective of developing a modified attenuate vaccine against MERS-Cov there are different points which deserves further characterization or at least needed to be more detailed in the discussion sections:

1. The authors demonstrated that no reversion of V1691R mutation was observed over five passages in cell culture and 4 days post infection in mice. This is an important information but certainly limited. Prolonged follow-up, with serial passages in vivo and co-infection studies would be required in order to guarantee the long-term genetic stability of the vaccines or absence of risk of complementation by wild type viruses.
2. Circulation among individuals of the attenuated vaccine may be of concern, especially in populations with increased vulnerabilities, like immuno-compromised patients. This may be assessed in transmission studies in animal models.
3. Attenuation in mice was assessed at intermediate challenge doses. What would be the outcome in animals exposed to high doses of the DUB-neg vaccine (i.e; 10⁶ or 10⁸ pfu)?

Roger Le Grand

Response to reviewers; Point-by-point reply

This manuscript describes the development of a live-attenuated MERS vaccine candidate by one amino acid substitution (V1691R) of the viral protein PLpro rendering deubiquitinating enzyme (DUB) negative strain. While the mutation does not hinder viral growth in vitro, the mutant grows 2 log less in vivo and causes attenuated disease in hDPP4 mice. The virus is cleared in 14 days post infection and all animals survive with less lung pathology compared to the infection with wt rMERS-CoVMA. The authors also show that DUB-negative strain induces early innate immune activation in mice. Single intranasal immunization with the mutant elicits sterilizing neutralizing antibodies that prevent viral infection from lethal challenge. The authors indicate that the protection is largely due to neutralizing antibodies as passively transferred animals were also protected, though not fully.

The paper is well written, straightforward and the outcome data support the hypothesis and the conclusion. Here are my suggestions;

We thank the reviewer for his/her very positive evaluation and valuable suggestions.

1. To my understanding, there are 2 set of rMERS-CoV one is based on EMC/2012 and the other one is based on a mouse-adapted strain, indicated as rMERS-CoVMA. The rMERS-CoV were tested on human cell lines and the mouse-adapted strain were tested on the mice. However, it is not clearly indicated that the DUB negative tested on mice model was rMERS-CoVMA-DUB. It was only indicated in the figures but missing indication of MA in the text. Therefore, it is a bit confusing and can be interpreted that the immunized DUB-negative has genetic background of the wt cell-adapted and not from mouse-adapted strain. An example:

Line 111: rMERS-CoV and the rMERS-CoVdubneg (DUB-negative MERS-CoV) then in line 115-114 rMERS-CoVMA and DUB-negative MERS-CoV is the same as non-mouse-adapted strain. It will be clearer to readers if DUB-negative MERS-CoV with genetic background from mouse-adapted strain are indicated as DUB-negative MERS-CoVMA where applicable.

The reviewer is correct, indeed there are two sets of rMERS-CoVs used in this study, one based on the EMC/2012 isolate and the other one based on a mouse-adapted MERS-CoV. All the recombinant viruses used in this study including the rMERS-CoV and rMERS-CoV_{DUBneg}, and the mouse-adapted viruses (rMERS-CoV_{MA} and rMERS-CoV_{MA-DUBneg}) were tested in human cell lines, see **Fig. 1A, B and Supplementary Fig. 1A, B**. The mouse-adapted viruses were further tested in human DPP4 KI mice. We thank the reviewer for pointing out that we should have made the textual distinction between these viruses more consistent/clear. To improve clarity, we have now changed the “DUB-negative MERS-CoV” with the genetic background from a mouse-adapted strain to “DUB-negative rMERS-CoV_{MA}”, where appropriate in the text.

2. Only five passages are not enough to demonstrate genetic stability of a virus intended for use as a vaccine. Most LAV studies have performed at least P10, which is usually required in vaccine manufacturing process. Also, it was not clear whether only the region of the substitution (V1691R) was sequenced or the whole genome. The NGS data should be included in the supplementary figure to clearly demonstrate that there are no mutations in P5 and P10 both in substitution region and elsewhere in the genome that could affect the phenotype of the virus.

We agree with reviewer 1 that five rounds of passaging in cell culture is limited for a virus intended for use as a vaccine. As discussed in the manuscript, our studies provide a proof-of-concept for the design of MLV coronavirus vaccines based on the selective inactivation of their PLpro DUB activity. Further development of the attenuated DUB-negative MERS-CoV would indeed have to include an extensive analysis of the genetic stability of the mutant virus in both cell culture (for manufacturing

purpose) and *in vivo* (for safety assessment). The incorporation of some of the strategies discussed in the manuscript might further improve the safety profile and stability of a candidate live vaccine. In particular, a combination of DUB-inactivating mutations could be developed to minimize the problem of (pseudo)reversion, but we consider such studies clearly beyond the scope of the present manuscript.

However, in order to explore reviewer 1's concern, and develop a first impression of the genetic stability of the introduced mutation in cell culture, we have now passaged both the wt (rMERS-CoV) and the DUB-negative rMERS-CoV in Huh7 cells 10 times. A summary of the NGS data is now included in **Supplementary Table 1**. We have adjusted the text in the Results/discussion section to accommodate this work and the conclusions. The NGS analysis of the full genome of the DUB-negative rMERS-CoV P10 virus indicate that 62% of the sequences still show the originally introduced DUB-inactivating mutation, and the variations seen at the mutated codon were predominantly substitutions to C, S or H, each requiring a single nt substitution (GTG = Val; CAC = His; AGC = Ser and TGC = Cys) and each occurring with a frequency of around 10%. Importantly, full reversion to a V codon (requiring 2 nt substitutions) could not be detected. We also found some low frequency mutations (mostly below 10% of the population) in other regions of the viral genome sequence.

3. In this study, the role of ORF5 in reducing NF-KB was not mentioned, as the genetic background of all of rMERS-CoVs used here were from the cell-adapted EMC/2012 isolate and contain premature stop codon 108 in ORF5. The effect of DUB-negative virus in animal model observed here might not translate to human, especially knowing that complete deletion of ORF5 in MA strain increases virulence in mice (reference 36. Gutierrez-Alvarez, J., et al 2021)

A premature stop codon 108 in ORF5 was deliberately introduced in all rMERS-CoVs used in this study to avoid complications during virus passaging due to ORF5 evolution and associated changes in host immune suppression in cell culture systems and mouse lungs (reference 34, 35, 36, Menachery, V.D, et al 2017 and also mentioned and discussed in **Materials and Methods**).

While the absence of ORF5 may enhance pathogenesis in mice (reference 36), the mouse-adapted parental virus (rMERS-CoV_{MA}) used to generate the DUB-negative virus causes a lethal lung disease in mice while the DUB-negative virus is strongly attenuated.

ORF5 has been reported to be highly stable *in vivo* in both human and camel isolates (reference 36). Future studies in camels utilizing wt and DUB-negative viruses that express full-length ORF5 or which have a complete deletion of ORF5 or a premature stop codon 108 in ORF5 (used in this study), might shed light on whether the effect of the DUB-negative virus seen in this mouse model can translate to camels, where ORF5 - as in humans - seems to be stable. Since we think this is beyond the scope of this work, we have not further elaborated on it in the manuscript. We did add a note on ORF5 in the discussion to address this (reviewer's point 12), please see below.

4. An evidence of T-cell response stimulated by the DUB-negative mutant is missing in the manuscript. The passive immune transfer failed to afford complete protection, suggesting that T-cell responses also play an important role.

We thank the reviewer for this useful suggestion and have now performed an additional animal experiment to test the effect of DUB-negative rMERS-CoV_{MA} vaccination on T-cell immunity in hDPP4 KI mice, while comparing with mock-vaccinated mice. Using an IFN- γ ELISpot assay, we measured splenic T-cell responses against a pool of peptides spanning the complete spike protein sequence (see **Material and Methods**). At four weeks post-vaccination, mice immunized with the DUB-negative rMERS-CoV_{MA} elicited significantly higher levels MERS-CoV spike specific IFN- γ producing T cells compared to the mock vaccinated animals. This data shows that the DUB-negative virus elicited S-specific cellular responses in mice (**Supplementary Fig. 6**) and in the **Discussion** suggest that T-cell

responses together with neutralizing antibodies may indeed also play an important role in protection against MERS-CoV.

5. Line 134: the sentence “DUB-negative MERS-CoV was cleared at day 6” is not correlated with the graph as there were still 103 viral titers.

The original sentence did not state that the “ DUB-negative MERS-CoV was cleared at day 6”. Please see the original sentence below. However, to avoid any confusion, we have now modified the sentence as shown below.

Original sentence: Furthermore, DUB-negative rMERS-CoV_{MA} was cleared from the lungs and at day 6 p.i. lung virus titers for the modified virus had significantly decreased to $\sim 1 \times 10^3$ PFU per g of lung tissue for 50% of the animals, while no progeny was measured for the other 50% of the animals at that time point (**Fig. 1D**).

Changed sentence: Furthermore, over time the DUB-negative rMERS-CoV_{MA} was cleared from the lungs. At day 6 p.i., lung virus titers for the modified virus had significantly decreased to $\sim 1 \times 10^3$ PFU per g of lung tissue for 50% of the animals, while no progeny was measured for the other 50% of the animals at that time point (**Fig. 1D**).

6. Line 248: 11 weeks is not quite qualified as long-lasting. I suggest to use “sustained” neutralizing antibodies to 11 weeks.

Where appropriate in the text, we have changed the term “long-lasting” to “sustained” as suggested by the reviewer.

7. Fig 6D: please indicate the limit of detection on the graph.

We have included the limit of detection for infectious viral progeny titers, which is now indicated with a dashed line. See **Fig. 6D** and also where relevant for other figures.

8. Line 275, 279, and 281: There is no lung pathology pictures included in supplementary Fig. 4. It is important to see overall lung pathology of immunized animals rather than just the semi-quantitative lung pathology score as it can be subjective, particularly when DUB-negative strain itself gave some lung pathology to the immunized animal (figure 3).

We have used the semi-quantitative lung pathology scores aiming to capture the severity and extent of the observed lung lesions in an unbiased and reproducible manner across all animals from both experiments (as explained in the methods section). Nevertheless, we agree with reviewer 1 that it may help to add some photomicrographs of lungs with representative lesions in **Supplementary Fig. 5**, to better appreciate the differences between mock and DUB-negative rMERS-CoV_{MA} immunized animals.

We have now added two such photomicrographs in **Supplementary Fig. 5A** and moved the bar chart to **Supplementary Fig. 5B**. To accommodate this change further in the Results section, we have also adjusted the text.

9. Line 299: As previously mentioned, T cell responses data should give a clearer picture to the absence of full protection after passive transfer. It should be interesting to know how T-cell responses of a live attenuated vaccine compares to viral vectored MERS vaccines, MVA and ChAdOx1 MERS vaccines.

As mentioned in the response to the previous comment (No. 4), we performed an additional animal experiment to evaluate whether the DUB-negative rMERS-CoV_{MA} is capable of inducing cellular

responses by 4 weeks post vaccination. Our data show that a single-dose of the DUB-negative rMERS-CoV_{MA} induces cellular responses in mice (**Supplementary Fig. 6**). We agree with reviewer 1 that it will be very interesting to learn how the T-cell responses induced by a live attenuated vaccine, like the DUB-negative rMERS-CoV_{MA}, compare to those induced by virally vectored MERS vaccines, like the MVA and ChAdOx1 MERS vaccines. However, in our opinion, such an elaborate comparison is clearly beyond the scope of this study, and will need to be the focus of future experiments aiming to compare the (humoral and cellular) immunogenicity of these vaccines side by side in the same animal model. As mentioned in the Discussion section of the original manuscript, live attenuated vaccines often induce excellent immune responses (humoral and cellular) and often provide lifelong immunity. Similar to a natural infection with MERS-CoV, live attenuated MERS-CoV vaccines are expected to induce broad T cell and humoral immune responses. The ability to intranasally deliver live attenuated vaccines like the DUB-negative MERS-CoV may provide a major advantage due to enhanced mucosal immunity compared to other vaccines, including MERS-CoV vectored candidate vaccines, which are mostly administered intramuscularly. Live attenuated vaccines also induce immunity against a range of MERS-CoV proteins, in addition to the spike proteins used in vectored vaccines, thus providing additional viral epitopes.

10. Line 379: It is highly likely (Chose one but as Nature Comm prefers us not to use highly or very. Maybe go with Likely or another synonymous word.)

We have taken reviewer 1's feedback and have changed "highly likely" to just "likely" in the text and where fitting elsewhere in the manuscript.

11. Line 429-434: There is not enough data to state that "the mutation introduced in MERS-CoV PLpro has high stability against possible reversion" as the virus can mutate elsewhere in its genome and recover its virulence. Line 431; no NGS data of at least passages 10 to back up the statement.

We agree with the reviewer that there is not enough data to state that "the mutation introduced in MERS-CoV PLpro has high stability against possible reversion". We have explored reviewer 1's concern about the genetic stability of the introduced mutation in cell culture, where both the wt rMERS-CoV and the DUB-negative rMERS-CoV viruses were passaged 10 times in Huh7 cells and analyzed by NGS (see also our response to point **No. 2**). It is important to stress that after 10 passages, and in spite of some evolution occurring at the mutated codon 838/1691, full reversion had not occurred and no variants have emerged that have become dominant in the population, with most abundant (non-synonymous) single-nt variations observed not exceeding 20.3% (Data included in **Supplementary Table S1**).

We have now modified the text in our Discussion and changed our statement to reflect the initial results from the full-genome sequencing of the P10 viruses (wt rMERS-CoV and the DUB-negative rMERS-CoV p10).

12. The deletion of accessory proteins is to be cautioned as the previous MA strain that was engineered with full deletion of ORF5 was more virulent in mice than the early stop codon mutant. TRS mutation maybe safer in terms of reducing recombination events and improving the stability of the engineered virus.

We agree with reviewer 1 that the deletion of accessory proteins is to be cautioned as the previous mouse adapted virus engineered with full deletion of ORF5 was more virulent in mice (reference 36). We have now made that clear in our discussion and have discussed the re-engineering of viral transcription regulatory sequences as a strategy to reduce the likelihood of successful recombination with other coronaviruses other than MERS-CoV and improving the stability of the engineered virus.

13. Line 470-478: The protocol for the construction of rMERS-CoV is not quite clear about the genetic background of the virus used. Please consider rephrasing. Also, where applicable, briefly describe short protocols rather than just quoting references.

Recombinant rMERS-CoV and rMERS-CoV_{DUBneg} were derived from a MERS-CoV full-length cDNA clone based on MERS-CoV strain EMC/2012 (reference 73). The recombinant mouse-adapted viruses rMERS-CoV_{MA} and rMERS-CoV_{MA-DUBneg} were generated from the pBAC-MERS^{FL}-MA-5FL full-length cDNA clone (reference 36) that was based on the mouse-adapted MERS_{MA}6.1.2 virus (reference 35). As Nature Communications requires that the Methods section be written as concisely as possible, we have now added the information about the genetic background of the viruses used in **Materials and Methods** by including the references in which they have been extensively described.

14. Line 741-743 and line 830-832: The reference numbers 34 and 35 are the same as reference numbers 71 and 72.

We apologize for this mistake and have now deleted reference numbers 71 and 72, and adjusted the referencing in the text.

15. Line 850: growth curve figure 1A and 1B were performed only once.

The growth curve in **Fig. 1A and 1B** was performed twice (now also made clear in the figure legend), however the replication of all viruses used in this study were evaluated at specified time points at least three times and statistical comparisons performed (see also **Supplementary Fig. 1**).

16. Figure 3: Consistency of the order of the data demonstrated; rMERS-CoV_{MA-DUBneg} (in red) was shown after rMERS-CoV_{MA} (in blue) in previous figure 1 and 2. Changing the order to be the same as the previous figure will also correlate with fig 3A as well.

We apologize for this inconsistency of the order of the data and where feasible in all the figures, we have now kept the order of the rMERS-CoV_{MA-DUBneg} (shown in red) and the wt rMERS-CoV_{MA} (shown in blue) consistent.

17. Figure 5B: The immunization is a single intranasal administration of the DUB-negative strain. Please explain why the neutralizing antibodies were increased at week 7 while the animals were not boosted.

Yes, the immunization is a single intranasal administration of the DUB-negative strain, however as the reviewer has pointed out the neutralizing antibody levels appear to be increased at week 7 while the animals were not boosted. The neutralizing antibody response seems to increase from week 2 to 4, then remain more or less constant between week 4 and 11 (**Supplementary Fig. 4**). The neutralization assays/runs were performed independently for each specified time point post vaccination and not in the same experiment/run. So the modest increase at week 7 (**Fig. 5B**) and the drop at week 9 (**Supplementary Fig. 4**) could be a result of experiment-to-experiment variation and do not necessarily reflect a change in neutralizing antibody levels in the vaccinated animals. We did not have a known standard neutralizing antibody with a known/established titer to control our experiments unfortunately. We did however include the virus back titration in each run to confirm the dose of virus inoculum. The assay used is very sensitive (IC₁₀₀), CPE-based and less effective in providing more quantitative measurements of the antibody strength. The acceptable back titration range of 30 to 300 TCID₅₀ is also quite wide. While we targeted to use a dose of 120 TCID₅₀/60 µL MERS-CoV, the back titration titer of the dose used in week 7 was 55 TCID₅₀/60 µL compared to 190 TCID₅₀/60 µL MERS-CoV in week 9, which might explain measuring increased neutralizing antibody

levels at week 7 and decreased neutralizing titers at week 9. Due to the limited amounts of sera recovered from immunized animals for each time point, it was not possible to repeat the assay for all the samples (from different time points) in one single run for direct comparison.

Moreover, it was necessary to perform the experiments for specified time points in independent runs as some time points before challenge with the lethal rMERS-CoV_{MA} (week 4 to 7) served as Go/No Go decision determinants requested by our animal welfare department.

We have now added a note in our discussion to address the reviewer's concern.

18. Figure 6D shows complete sterilization of the inoculum virus as no infectious particles are detected in the lungs. However, the authors should perform qRT-PCR to determine the level of genomic viral RNA, which could give insight on viral inhibition in immunized animals.

We have now performed the requested qRT-PCR analysis to determine the levels of genome (ORF1a) and sub-genomic (N gene) viral RNA in the lungs of DUB-negative rMERS-CoV_{MA} vaccinated mice at day 0, 2, 4, 6 and 14 after challenge with a lethal dose of rMERS-CoV_{MA} (Materials and Methods). On day 2 and 4, the lungs of mock-vaccinated animals had high MERS-CoV RNA levels (**Supplementary Fig. 7**) consistent with the high levels of infectious virus (**Fig. 6D**). In contrast, DUB-negative rMERS-CoV_{MA}-vaccinated mice had significantly reduced viral RNA levels (at least 5 log reduction) in the lungs at day 2 and 4 post challenge, which were similar to viral RNA levels at day 0, 6 and 14 (**Supplementary Fig. 7**). This data is consistent with the undetectable levels of infectious virus in the lungs of vaccinated animals (**Fig. 6D**) demonstrating the complete sterilization of the lethal dose of wt virus in DUB-negative rMERS-CoV_{MA} immunized animals.

Reviewer #2 (Remarks to the Author):

Myeni et al describe *in vitro* and *in vivo* properties of an attenuated construct of MERS-Cov they propose as a new candidate for a human vaccine. Previous work demonstrated that substitution in the ubiquitin binding site of MERS-Cov papain-like protease (PLpro) disrupt its deubiquitinating enzyme (DUB) activity. Authors confirm that this modification is not affecting viral replication dynamics in Huh7 and MRC5 human cell lines. However, the V1691R mutation abrogating DUB activity also result in reduced IFN- β promoter inhibition therefore restoring host cells innate response capacity. The attenuated phenotype of the DUB-negative MERS-Cov is confirmed in the DPP4 knock-in mouse model highly susceptible (100% mortality in about 5 days post-infection) to mouse adapted MERS-Cov. The V1691R attenuated virus have reduced replication capacity in the lung of challenged mice and accelerated clearance. In addition, mice infected with DUB-neg attenuated virus have significantly reduced lung pathology and none of the animals died from infection. Interesting, at day 1 pi, attenuated phenotype in mice is associated with increased Type I and Type III interferon response as well as increased IL-6 and TNF- α , in agreement the impaired capacity of the attenuated virus to counteract innate host response. Finally, a single intranasal immunization of mice with the DUB-negative attenuated virus (10⁴ pfu) induce strong neutralizing antibody response persisting up to 9-11 weeks. These nAb have broad activity against a diversity of MERS-Cov strains. Sterilizing immunity was obtained against lethal MERS-Cov challenge and passive transfer study in mice demonstrated that neutralizing antibodies contribute to a significant part of observed protection.

To conclude, this is a very well described study with strong methodology convincing on the attenuated phenotype of the DUB-negative virus in cell cultures and in a the mice model. However, in the perspective of developing a modified attenuate vaccine against MERS-Cov there are different points which deserves further characterization or at least needed to be more detailed in the discussion sections:

We thank the reviewer for his very positive evaluation and we appreciate the valuable suggestions.

1. The authors demonstrated that no reversion of V1691R mutation was observed over five passages in cell culture and 4 days post infection in mice. This is an important information but certainly limited. Prolonged follow-up, with serial passages *in vivo* and co-infection studies would be required in order to guarantee the long-term genetic stability of the vaccines or absence of risk of complementation by wild type viruses.

Please refer to **Reviewer #1, point 2 and 11**. We also agree with reviewer 2 that ideal future experiments (beyond the scope of this study) will have to look into prolonged follow-up serial passages *in vivo*, NGS analysis of p10 or higher viruses from *in vivo* passaging with statistical replicates, and the incorporation of strategies discussed in the manuscript to improve the safety profile and stability of the live attenuated DUB-negative virus to guarantee the long-term genetic stability of the DUB-negative MERS-CoV.

2. Circulation among individuals of the attenuated vaccine may be of concern, especially in populations with increased vulnerabilities, like immuno-compromised patients. This may be assessed in transmission studies in animal models.

We agree with reviewer 2 that the circulation of modified live attenuated viruses like the DUB-negative MERS-CoV may be of concern in populations with increased vulnerabilities. Live attenuated vaccine candidates like the DUB-negative virus administered intranasally might however, have an advantage of priming early protective innate cellular responses as well as inducing localized adaptive

memory through subsequent infection of the upper respiratory tract (also mentioned in the **Discussion**). As a result, this localized adaptive memory would significantly reduce/abrogate disease and symptoms therefore prevent or considerably reduce transmission to high-risk populations or immunocompromised patients. This of course would have to be assessed in the future in appropriate animal transmission models for MERS-CoV. Alternatively, the DUB-negative rMERS-CoV could be engineered to allow for single cycle replication without spread in a trans-complementation system (replicon RNAs in cells that express missing genes like structural genes in the replicon), recently demonstrated for SARS-CoV-2 by Inna Ricardo-Lax, et al Science, 2021.

3. Attenuation in mice was assessed at intermediate challenge doses. What would be the outcome in animals exposed to high doses of the DUB-neg vaccine (i.e; 10^6 or 10^8 pfu)?

While it would be interesting to explore the influence of high doses of the attenuated DUB-negative virus in mice, this was not possible because both the wt rMERS-CoV_{MA} and the DUB-negative rMERS-CoV_{MA} viruses grew to similar but lower titers (between 5.2×10^6 to 8.8×10^6 PFU/mL) in human and primary mouse cell cultures. Thus it was not feasible to get to 10^6 or 10^8 PFU/50uL intranasal dose (maximum volume allowed for intranasal administration in mice is 50uL). It is important to emphasize that the doses (10^4 to 10^5) used in this study of the wt rMERS-CoV_{MA} virus are highly lethal in mice (see **Supplementary Fig. 2**). An inoculum of 5×10^3 PFU was established as the lethal dose in hDPP4 KI mice (reference 35). It is also important to emphasize that even though high infectious virus loads of the DUB-negative rMERS-CoV_{MA} virus ($>10^6$ PFU/g lung) were recovered after 1-2 days of infection in mice, all animals infected with the DUB-negative virus survived and, like the mock-infected animals, showed no signs of morbidity and kept a relatively stable body weight (see also **Fig. 1 and Fig. 2**).

Reviewer Comments (Remarks to the Author)

The Reviewer would like to thank the authors for considering the points made and the extra data provided which enhance the clarity of the content of this manuscript. Most points were clearly addressed, only a couple of small questions that may be discussed.

1. L127: V383C,S,H mutants would still render the same DUB-negative phenotype?
2. L297: A typo of (“) Before -Taken together.
3. Supplementary figure 6A and B were done from the same samples used in figure 6. The viral RNA detected $\sim 10^{11}$ or 10^{12} copies were correlated to $\sim 10^7$ - 10^8 infectious particles however when the viral RNA detected at 10^6 RNA copies, this does not confer at least 10^1 - 10^2 particles which would be within the limit of detection (10 PFU). Would you maybe discuss or speculate on this discrepancy?

Response to reviewers; Point-by-point reply

The Reviewer would like to thank the authors for considering the points made and the extra data provided which enhance the clarity of the content of this manuscript. Most points were clearly addressed, only a couple of small questions that may be discussed.

We would like to thank the reviewer for his/her comments and suggestions to further improve the manuscript.

1. L127: V383C,S,H mutants would still render the same DUB-negative phenotype?

This is an interesting question from the reviewer that requires further investigation that we think is beyond the scope of the present manuscript. In this work, we have aimed to develop a first impression of the genetic stability of the mutant virus in the substitution region or elsewhere in the genome. Additional stability experiments including statistically relevant replicates in primary cell culture systems and prolonged follow-up serial passages *in vivo* would need to be performed in order to conclude on the validity of the V383C,S, H mutants. As stated in the manuscript, after 10 passages, reversions to the wild-type V residues (requiring a double nucleotide substitution) could not be detected, demonstrating that modified rMERS-CoV carrying a change in the Ub-binding site of PLpro was viable and reasonable stable in cell culture. Our studies provide a proof-of-concept for the design of MLV coronavirus vaccines based on the selective inactivation of their PLpro DUB activity. Further development of the attenuated DUB-negative MERS-CoV would need to include extensive analysis of the genetic stability the mutant virus. As discussed in the manuscript, a combination of DUB-inactivation mutations could be developed to minimize the problem of (pseudo)reversion, but we consider such studies clearly beyond the scope of the current manuscript.

2. L297: A typo of (“ Before -Taken together.

We apologize for this typo and thank the reviewer for pointing this out and have deleted the (“ before “Taken together” in the sentence.

3. Supplementary figure 6A and B were done from the same samples used in figure 6. The viral RNA detected $\sim 10^{11}$ or 10^{12} copies were correlated to $\sim 10^7$ - 10^8 infectious particles however when the viral RNA detected at 10^6 RNA copies, this does not confer at least 10^1 - 10^2 particles which would be within the limit of detection (10 PFU). Would you maybe discuss or speculate on this discrepancy?

It is important for us to first clarify to the reviewer that the viral RNA copies and PFUs are not from the same sample but from the same animal and different parts of the lung with possible different viral distribution. Different lung homogenization protocols were followed for measuring viral RNA copies and PFUs as stated in the Materials and Methods. The lungs from each animal were sectioned into different parts for various outcome parameters including the viral RNA copies, PFUs/cytokine measurements and pathology. To evaluate viral RNA copies, the day 0 samples (no virus) served as the reference for the viral RNA copies, which were also normalized to the lung weight and a housekeeping gene. We have now also included the limit of detection for viral RNA copies, which is indicated with a dashed line. See Supplementary Figure 7A (viral sub-genomic RNA) and B (genomic RNA). The level of genomic viral RNA clearly demonstrates viral inhibition in the immunized animals compared to the mock-immunized animals, which is consistent with no infectious viral particles detected in the lungs. Moreover, viral RNA detection has been shown not to necessarily correlate with infectiousness in animal models. *Nature* **583**, 834–838 (2020) and *Nat. Commun.* **12**, 267 (2021).